# Stress-Responsive Gene Expression, Metabolic, Physiological, and Agronomic Responses by Consortium Nano-Silica with Trichoderma against Drought Stress in Bread Wheat

**DOI:** 10.3390/ijms252010954

**Published:** 2024-10-11

**Authors:** Ghalia S. Aljeddani, Ragaa A. Hamouda, Amal M. Abdelsattar, Yasmin M. Heikal

**Affiliations:** 1Department of Biology, Collage of Science, University of Jeddah, Jeddah 21589, Saudi Arabia; gsalgdanee@uj.edu.sa; 2Department of Applied Radiologic Technology, College of Applied Medical Sciences, University of Jeddah, Jeddah 23218, Saudi Arabia; ragaahom@yahoo.com; 3Department of Microbial Biotechnology, Genetic Engineering and Biotechnology Research Institute (GEBRI), University of Sadat City, Sadat City 32897, Egypt; 4Botany Department, Faculty of Science, Mansoura University, Mansoura 35516, Egypt; amal_am1994@std.mans.edu.eg

**Keywords:** plant growth-promoting fungi, rice husk, green synthesis of nanoparticles, KEGG pathway, *Triticum aestivum* L., drought-responsive genes

## Abstract

The exploitation of drought is a critical worldwide challenge that influences wheat growth and productivity. This study aimed to investigate a synergistic amendment strategy for drought using the single and combined application of plant growth-promoting microorganisms (PGPM) (*Trichoderma harzianum*) and biogenic silica nanoparticles (SiO_2_NPs) from rice husk ash (RHA) on Saudi Arabia’s Spring wheat Summit cultivar (*Triticum aestivum* L.) for 102 DAS (days after sowing). The significant improvement was due to the application of 600 ppm SiO_2_NPs and *T. harzianum* + 600 ppm SiO_2_NPs, which enhanced the physiological properties of chlorophyll a, carotenoids, total pigments, osmolytes, and antioxidant contents of drought-stressed wheat plants as adaptive strategies. The results suggest that the expression of the studied genes (*TaP5CS1*, *TaZFP34*, *TaWRKY1*, *TaMPK3*, *TaLEA*, and the wheat housekeeping gene *TaActin*) in wheat remarkably enhanced wheat tolerance to drought stress. We discovered that the genes and metabolites involved significantly contributed to defense responses, making them potential targets for assessing drought tolerance levels. The drought tolerance indices of wheat were revealed by the mean productivity (MP), stress sensitivity index (SSI), yield stability index (YSI), and stress tolerance index (STI). We employed four databases, such as BAR, InterPro, phytozome, and the KEGG pathway, to predict and decipher the putative domains in prior gene sequencing. As a result, we discovered that these genes may be involved in a range of important biological functions in specific tissues at different developmental stages, including response to drought stress, proline accumulation, plant growth and development, and defense response. In conclusion, the sole and/or dual *T. harzianum* application to the wheat cultivar improved drought tolerance strength. These findings could be insightful data for wheat production in Saudi Arabia under various water regimes.

## 1. Introduction

In terms of producing grains for the human diet, wheat is among the most significant cereal crops in the world [1]. Around 36% of people on the planet use wheat as their primary staple food, which provides 20% calories and 55% carbohydrates [2]. After rice (*Oryza sativa* L.) and maize (*Zea mays* L.), wheat is the third most-grown crop in the world, with an estimated yearly production of 600 million metric tons [3]. Currently, *Triticum aestivum*, or bread wheat, accounts for around 95% of the world’s crop production [4]. Environmental stresses have significant consequences for yield. Environmental stresses have become the primary challenge for the production of staple crops in a changing climate. The growing global warming and depletion of water supplies, together with the deteriorating state of the environment, have had a negative impact on wheat production recently. This has imperiled the growing population’s access to adequate nourishment [5].

Water shortage, or drought stress, throughout the growth season, as well as general stress in most arid and semi-arid environments, is among the environmental factors significantly lowering productivity [6]. The UN has identified Saudi Arabia as a country suffering a scarcity of water [7]. The Water Resources Institute estimates that by 2040, Saudi Arabia will rank ninth globally in terms of water stress [8]. The region is particularly sensitive to the detrimental impacts of climate change because of the country’s inherent aridity and water shortages brought on by climate change [9]. Even a one-degree-Celsius rise in average temperature can significantly affect the amount of water needed for agricultural irrigation, resulting in a 5-to-25% reduction in yields of different crops grown in Saudi Arabia [10]. In Saudi Arabia, wheat cultivation is given more consideration. However, Saudi Arabia’s wheat growing has been restricted because of the country’s limited water supplies. Much of the land in the Kingdom of Saudi Arabia is unsuitable for agriculture due to topography, little rainfall, and excessive salinity [11].

It is expected that by 2025, there will be 8.3 billion people in the entire world, increasing from the present 7 billion. The globe will need 70–100% more food by 2050 [12]. Nowadays, the bulk of soil management practices uses fertilizers derived from inorganic chemicals, which puts the environment and general public’s health at a high risk [13]. Drought has a negative effect on several plant morphological and physiological characteristics, such as relative water content, leaf water potential, plant height, leaf area, and chlorophyll levels [14,15]. Drought stress negatively affects wheat production. When there is 40% less water, Daryanto, et al. [16] reported yield reductions of 20.6%. The most vulnerable growth stage to water deficit for wheat yield is from double-ridge to anthesis because of the detrimental effects on the quantity of the spikelets and, eventually, kernels per spike.

Wheat cultivars that grow even in harsh environments can be developed through drought mitigation methods like resource allocation; molecular breeding; seed, bacterial, fungal, and hormonal priming; physiological trait-based breeding; water budgeting; and a combination of these techniques [17]. Plants have developed a variety of sophisticated defense mechanisms over time to withstand environmental stressors such as drought stress via a variety of physiological, morphological, metabolic, and molecular processes [18].

Seed priming is an easy and effective hydration technique to encourage seed germination [19]. It increases crop yield, the intake of nutrients, efficiency of water usage, emission of photo- and thermo-dormancy, and the activation of stress-responsive genes and proteins [20]. Based on the materials or solutions used, there are various forms of seed priming that are used, including living things (bio-priming), nanoparticles (nano-priming), etc. [21]. Firstly, a method for treating seeds (seed priming) called “seed bio-priming” is used to control plant development, manage stress, and enhance seed germination [22]. Seeds primed with plant growth-promoting microorganisms (PGPMs) which surround a plant’s root zone can greatly improve plant performance, even in unfavorable environments [23]. These endophytes are important colonizers, particularly in harsh environments where they encourage drought tolerance and boost plant development under stressful conditions [24,25]. Among these helpful microbes, the fungus Trichoderma has been investigated in great detail for its ability to stimulate plant development, generate abiotic stresses, and act as a biological control agent against plant diseases in a variety of plant species, including rapeseed [26,27]. Trichoderma species are often found in soil, and most strains have shown capacity to colonize root systems and form symbiotic connections with plants [28,29], becoming endophytes. According to some research results, Trichoderma improve plants’ ability to tolerate abiotic stress [30,31]. Trichoderma strains can release phytohormones, including cytokinins (CK), salicylic acid (SA), gibberellins (GA), auxins, and abscisic acid (ABA). These phytohormones influence processes like host plant growth, colonization, and the activation of defenses in stressful environments, and they add complexity to interactions between Trichoderma and plants [32].

Secondly, nanopriming through the addition of nano-enhanced solutions can improve plant defense, growth and development, and seed germination rates. This is primarily made possible by the careful or regulated dispersion of agrochemicals, which lowers the losses of nutrients via leaching, volatilization, and other processes [33]. Green synthesis employing eco-friendly agricultural waste silica nanoparticles was broadly investigated. Agricultural waste materials may be used to make rice husk; each year, more than 80 million tons of RHA (rice husk ash) is generated [34,35]. Reports state that it is lightweight, reproducing almost 0.23 tons of husk for each ton of rice utilized [36]. Since rice ash is readily available and low-cost, it provides researchers with important material, particularly on this continent, where numerous compositions are found [37].

Applying SiNPs to plants may successfully boost their capacity for photosynthesis, growth, and stress tolerance [38]. Si can reduce the impact of drought conditions on photosynthetic pigments through raised endogenous cytokinin levels, which restore chlorophyll synthesis and enhance chloroplast ultrastructure [39]. The use of nano-silica increases plant hydration status, boosts growth and productivity, and causes significant modifications in the ultrastructure of leaf organelles. It has also triggers the plant defense mechanism and improves the hunt for clear particles [40]. Furthermore, SiO_2_NPs have been identified as a plant growth inducer that increases root endodermal silicification, strengthens the antioxidant system during stress, and enhances the cell water balance and resistance to abiotic challenges [41]. Compared to untreated seeds, nano-priming-enhanced photosynthetic parameters, preserved biochemical balance, and enhanced biomass products in wheat seedlings [42].

Thirdly, recent advances in drought tolerance research have led to the discovery of a plethora of essential genes and transcription regulators that affect morphophysiological characteristics. To increase wheat’s resistance to drought, molecular breeding techniques have focused on genes that govern stomatal growth and root architecture because they are crucial for both retaining and extracting soil moisture. Recent advancements in wheat genomics and reverse genetics, such as the establishment of genome editing technologies and the availability of a gold-standard reference genome sequence, are expected to aid in identifying the functional roles of the genes and regulatory networks underlying adaptive phenological traits and in the development of drought-tolerant cultivars [43].

The wheat EST database contains cDNA library sequences from a variety of tissues, including cell cultures, spikes, crowns, flower organs (lemma, anthers, pistils, and palea), leaves, seedlings, roots, stems, seeds at various stages of development, endosperm, and embryo. Furthermore, the same database contains EST sequences from the cDNA libraries of wheat plants exposed to biotic (powdery mildew, *Fusarium graminearum*, Hessian fly, stripe rust) and abiotic (salt, drought, high and low temperatures, aluminum) conditions [44].

This current study highlighted the progress made in the morpho-physiological, biochemical, and molecular traits in wheat to maintain yield stability under depleting water conditions through the following: (1) mitigation of the drought stress on wheat by using eco-friendly methods through application of grain bio-nanopriming: (2) assessment of the morphophysiological parameters of treated wheat plants; (3) estimation of the physiological and biochemical variations on the treated plant seedling; (4) identification of five drought-responsive genes (*TaP5CS1*, *TaZFP34*, *TaWRKY1*, *TaMPK3*, and *TaLEA*) that modulate this drought-adapting mechanism might help us understand the molecular basis of adaptive features; (5) evaluation of some agronomic-related yield traits and measurements of some effective drought indices on the wheat yield; (6) construction of a wheat electronic fluorescent pictograph that involves tissue-specific and developmental stage expression information for the studied genes in water-deficit conditions.

## 2. Results

### 2.1. Plant Growth-Promoting Traits of T. harzianum

The cultural characteristics were detected as *T. harzianum* colonies showed typical green growth and different sporulation shapes. *T. harzianum* is micro-morphologically characterized by the presence of septate filamentous hyphae, branched conidiophores, flask shaped phialides, and variably shaped green conidia (globose to subglobose) (Appendix A). The estimation of *T. harzianum* in different concentrations of polyethylene glycol (PEG): 0, 15, 20, 25, 30, and 35% (Appendix A). During the in vitro screening procedure, *T. harzianum* was assessed on a PDB medium without PEG (negative control, 0%) and proved to be viable 5–7 days following inoculation. Nonetheless, *T. harzianum* was able to develop in PDB at various PEG levels. The mycelial growth of *T. harzianum* examined decreased with an increase in the negative matric water potential. Growth was completely bushy (++++) at 0 and 15%, whereas hyphal growth was on a (+++) scale at 20 and 25%, (++) at 30%, and (+) at 35%.

While discrete hydrolytic halos independently generated surrounding the colonies, *T. harzianum* was able to produce amylase and cellulase but was unable to produce the protease extracellular hydrolytic enzymes, as shown in Table 1. The development of a clear zone surrounding colonies on starch agar with the addition of iodine served as an indicator of amylase, as shown in Appendix A. After adding the Congo red reagent, a clear zone surrounding the colonies indicated the presence of cellulase, as in Appendix A. Protease was detected by the formation of a clear zone surrounding the colonies grown on skimmed agar.

Furthermore, because *T. harzianum* was grown on nitrogen-free Jensen’s agar, it can fix nitrogen, as observed in Appendix A. Nevertheless, the absence of noticeable hydrolytic halo formation surrounding colonies indicated that *T. harzianum* lacked the ability to solubilize phosphate in the experimental conditions, as examined in Appendix A. The formation of a pink color indicated the presence of IAA (8.50 ± 0.59 μg/mL), and the amount of GA produced was measured spectrophotometrically (28.62 ± 1.76 μg/mL) (Table 1). Additionally, color changes from yellow to brown were used to detect ammonia; however, *T. harzianum* was unable to do so, as expressed in Table 1.

### 2.2. Synthesis and Characterization of Green Synthesized Silica Nanoparticles (SiO_2_NPs)

Figure 1 demonstrates the stepwise synthesis procedures and extraction of pure silicon dioxide from rice husk ash to gel product to pure nano-powder of SiO_2_ through the refluxing method. Structural, morphological, and optical characterizations were measured through different techniques as follows: UV, XRD, EDX, SEM, FT-IR, zeta potential, and TGA analysis, which showed the revelation crystal size in the nanoscale and the stability of silica nano-powder. The UV spectra analysis of silica nanoparticles showed that the wavelength was located at 235 nm, as observed in (Figure 2). The zeta potential of silica nanoparticles that was synthesized by rice husk as precursors. The results showed the zeta potential of silica nanoparticles had a negative charge (−26.3 mV).

EDX analysis of silica nanoparticles revealed the presence of three elements oxygen, sodium, and silicon with percentage weights of 75.01, 3.90, and 21.09%, respectively. The low amount of sodium denoted the low contamination of silica nanoparticles, and the high amount of oxygen denoted the nanoparticles were SiO_2_ nanoparticles as shown in Figure 2. The XRD diffraction patterns of silica nanoparticles were recorded at 2 Theta 22.10°, 27.55°, 31.84°, 45.58°, 54.026°, 56.637°, 66.32°, and 75.39°, which correspond to lattice plane 100, 110, 110, 200, 211, 211, 221, and 311, respectively. The results showed the sharp peaks of silica nanoparticles which demonstrated the silica nanoparticles were crystalline. Appendix A shows the intensity % of the peaks, which revealed the crystalline size of silica nanoparticles as illustrated in Figure 3. The major crystalline peak 100% was obtained at 2 Theta 31.846°, which represents the size of the silica nanoparticle at 67.86 nm.

In addition, FT-IR spectroscopy analysis of nano-silica particles showed that there are five absorbance peaks that were noticed at wavelengths 2114, 1637, and 1046 due to asymmetric vibration SiO-Si (siloxane group), symmetric vibration at 793, and 618 cm^−1^ attributed to the bending vibration of Si-O, as shown in Figure 4a. Moreover, the TGA analysis of nano-silica demonstrated the initial weight loss of 4% at 180 °C due to loss of water, and further slightly increased the loss of weight 5% at 462 °C, followed by 6% at 743 °C due to the degradation of organic compounds which capped nano-silica. So, the results displayed the stability of nano-silica at high temperatures, as observed in temperature (Figure 4b).

### 2.3. Drought Responses of T. aestivum at Different Stages

#### 2.3.1. Morphological and Phenotypic Responses of *T. aestivum* at the Heading Stage under Different Water Regimes

The results of morphological features revealed significant variations between treatments, as shown in Table 2 and Figure 5. These variations are critical for evaluating the effect of *T. harzianum* and nano silica individually or in dual applications as T0 (control with dist. water), T1 (*T. harzianum*), T2 (SiO_2_NPs (600 ppm)), and T3 (*T. harzianum* + SiO_2_NPs (600 ppm)) on the morphological and phenological characteristics of 65 DAS of *T. aestivum* under different water regimes. The cell plot assessed 18 morphological and phenotypic traits of treated *T. aestivum* under different water regimes (D0, control; 100% FC; D1, mild drought with 50% FC; D2, severe drought with 30% FC), as shown in Figure 5. Under the full irrigation 100% FC condition (D0), most of the phenotypic traits revealed a significant increase in all samples after T1 application. PL, SHL, LL, FWSH, DWSH, FWR, and DWR showed the lowest values after T2 application, while WCSH and WCR traits showed an elevation after T2 application. Under mild drought with 50% FC (D1) condition, there was an improvement in FWSH, DWSH, FWR, and DWR content after T2 application, while T3 application showed an elevation in FWSH, FWR, and water capacity (WCSH and WCR) traits. In addition, there was a significant increase in SH/R after T3 application. There was no observable change on PL, SHL, and RL through different applications. After T1 application, the results revealed an increase in plant height and root traits (R. no. and RW). Under severe drought with 30% FC (D2), all the morphological and phenotypic traits showed an observable decline after T0 application. After T1 application, a good improvement in water relations and plant length expressed in FWSH, FWR, DWSH, DWR, WCSH, and WCR, and a slight improvement in leaf traits.

#### 2.3.2. Physiological and Biochemical Parameter Response *T. aestivum* at the Heading Stage under Different Water Conditions

*T. aestivum* showed different physiological, biochemical, and metabolic performances under the influence of the four treatments under three different conditions in this experiment. Cell plot assessed 14 physiological and biochemical traits of 65 DAS *T. aestivum* under different water regimes as expressed in (Table 3 and Figure 6). The highest value of membrane electrolyte leakage EL was recorded as 51.14% after T0 application under D2 conditions, while the lowest ones (2.77 and 2.71%) after T1 application under D1 and D2 conditions, respectively. Moreover, the photosynthetic pigments (Chl a, Chl b, Card, and TP) showed the lowest content in T0 under D0 conditions. Chl a, Card, and TP traits revealed an elevation after T2 and T3 applications under D1 and D2 conditions, respectively.

Remarkably, drought stress enhanced the osmolytes synthesis and enzymatic and non-enzymatic antioxidants of treated *T. aestivum* after treatment application, according to the results in Table 4. T3 application showed the highest accumulation of total carbohydrate (Carb) content (349.31 and 393.01 mg/g) under D1 and D2 conditions, respectively. On the other hand, protein (ProT) had the maximum content (38.33 and 24.44 mg/g) after T2 application under both D0 and D2 conditions, respectively. Moreover, T3 and T2 applications showed the highest proline (ProL) values (6.28 and 5.20 mg/g) under both D1 and D2 conditions, respectively, compared to the control values. Additionally, the total flavonoid content (TFC) increased (6.57 and 63.67 mg/g) after T1 application under both regulated D0 and stressful D1 conditions, respectively. T2 hardly had a significant impact on TFC under D2 condition, while it showed an elevation after all other treatments applications. Likewise, total phenolic content (TPC) showed the highest value (33.40 mg/g) when plants were subjected to T1 application under D0 condition. TPC in wheat leaves increased in a significant way after T1 and T3 applications under different water conditions, as shown in Figure 6.

In addition, DPPH content increased to 63.67, 65.90, and 65.93% after T1, T3, and T0 under D1, D0, and D2 conditions, respectively. According to non-enzymatic antioxidants, catalase, peroxidase, and polyphenol oxidase (CAT, POD, and PPO) activities were remarkably increased to 6.53, 73.68 and 38.40 U/g, respectively, after T2 application under D0 conditions.

Water stress induced the maximum activities in CAT to 9.57 and 9.17 U/g after T1 and T2 applications under D1 and D2 conditions, respectively. Moreover, T2 application represented the highest values of POD and PPO activities as 73.68 and 38.40 U/g under D0 condition. In addition to, T3 showed high PPO activity at 34.44 U/g under D1 conditions. Severe drought stress (D2) increased PPO activity after different treatment applications, as observed in Figure 6.

#### 2.3.3. Multivariate Analysis and Inter-Correlation Based on Morpho-Physiological and Biochemical Traits at the Heading Stage: Principal Component Analysis (PCA) Biplot and Heatmap Partial Correlation

Principal component analysis (PCA), score plot, biplot, and heatmap of partial correlation of all 32 combined heading traits (morphological, physiological, and biochemical traits) of 65 DAS of *T. aestivum* under different water regimes. PCA was described by the first two PCs per cluster, as shown in Figure 7a,b. The first component PC1 was scored as 30.6% and the second component PC2 as 19.7% of the total variation. In the PCA score plot shown in Figure 7a, *T. aestivum* samples were grouped in three clusters according to the three corresponding water regimes (D0, D1, and D2). The first red-dashed cluster had *T. aestivum* under D0 condition, while the second green-dash cluster represented plants under D1 condition which intercalated with the last, blue-dash cluster, which had all the rest of the samples under D2 condition.

Figure 7b shows the biplot in which the system of the two first components, length of vector and cosine of angle, were utilized for the discrimination of *T. aestivum* plants under different treatments and water regimes. In the first red D0 cluster, the most frequent vectors (the longest) were WCR, WCSH, DWSH, and FWSH which showed strong positive correlation among themselves based on the morphological attributes of shoot and root as shown in the heatmap partial correlation, in Figure 7c. Moreover, leaf traits such as LA, LW, L. no., and LP were the most distinctive vectors in differentiation among the treated *T. aestivum* samples. According to the physiological and biochemical traits, ProT, POD, and PPO were the most significant vectors and had strong positive correlation as shown in Figure 7c. On the other hand, the least vectors were EL, TFC, Carb, and Chl b. The results revealed strong positive correlations in this cluster, such as LA and LW, EL and Carb, WCR and WCSH, L. no. and FWSH, SH/R and LP, and TFC and EL as shown in Figure 7b,c.

However, the second green cluster was controlled by PL, RL, SHL, DWR, FWR, and DPPH vectors under D1 condition. According to the vector’s correlations, strong positive correlations were recorded among PL and SHL and DWR traits. Likewise, LA was strongly positively correlated with LW, LA, and LP. On the other hand, RW and FWR and DPPH, and RL and DPPH and TPC had negative correlations with ProT, POD, PPO, and Chl a. Moreover, SH/R and RL parameters differentiated among the first red and the second green clusters, as shown also in the heatmap partial correlation plot Figure 7c. In the third blue D2 cluster, the most significant vectors were Card, Chl a, CAT, ProL, FWSH, and L. no. While the shortest vectors were EL, TFC, Carb, Chl b, and TPC. Strong positive correlations were found Card and Chla, ProL and CAT and EL and TFC, while the negative correlations were recorded among Card and Chl a, ProL and CAT with LA and LW, and EL and TFC, as observed in Figure 7b,c.

### 2.4. Expression Analysis: Quantitative RT-PCR of Five Drought-Responsive Genes

The five drought-responsive genes of 65 DAS of *T. aestivum* at the heading stage were expressed in a distinctive way under different water regimes, as illustrated in Figure 8. The genes were defined as *TaP5CS1*, *TaZFP34*, *TaWRKY1*, *TaMPK3*, *TaLEA*, and the wheat housekeeping gene *TaActin*, used as the internal reference gene. *TaP5CS1*, a proline biosynthetic gene and proposed as an ABA responsive gene, was expressed in Figure 8a. Under D0 conditions, *TaP5CS1* had hardly an impact on *T. aestivum* samples under different treatments. On the other hand, it showed a high expression by ~4.5 folds after T0 application under D2 conditions and followed by ~4 folds after T2 application under D2 conditions. Under D1 conditions, *T. aestivum TaP5CS1* expression reached the maximum (~2.70 folds) after T1 application followed by (~1.36 folds) after T3 application and ended by (~1.32 folds) after T2 application.

Under D0 conditions, *TaZFP34*, which is known as modulation of root growth genes, showed a significant expression in all treated *T. aestivum* by (~1 fold). Moreover, it overexpressed by ~2 folds with T3 application followed by ~1.7 folds with T1 application then by (~0.5-fold) by T2 application under D1 condition. On the contrary, it had no expression under D2 conditions, as observed in Figure 8b. In addition, *TaWRKY1*, an ABA signaling transduction gene, showed non-significant change among wheat samples under different treatments under D0 conditions. Under D2 conditions, different treatments did not affect gene expression, except after T0 application, as illustrated in Figure 8c. *TaMPK3*, mitogen-activated protein kinase, *MPK3*, showed no change in its expression among wheat samples under different treatments under D0 conditions, while *TaMPK3* recorded the maximum expression after T2 application in D1 conditions. In addition, its expression showed a little variation after T2 and T3 applications under D2 condition, as observed in Figure 8d. *TaLEA*, a late embryogenesis abundance protein, was expressed in the same manner in all treated wheat samples under D0. However, *TaLEA* was expressed (~9 folds) after T0 application under D1 conditions. Under severe drought (D2), *TaLEA* is overexpressed (~28 folds) after applications of T0 followed by T1.

### 2.5. Yield-Related Agronomic Attributes of T. aestivum

#### 2.5.1. Yield Traits with Multivariate Analysis

The agronomic attributes of 22 yield traits of 102 DAS of *T. aestivum* under different water regimes are shown in Table 4. Based on the shoot and root traits, the highest values of FWSH, FWR, DWSH, WCSH, and WCR were 1.56 g, 0.16 g, 1.27 g, 18.54%, and 40.95%, respectively, after T1 under D2 conditions, while the most minimal of those parameters were found after T2 under D0 conditions. In addition, PL and SHL recorded the longest length (59.72 and 49.29 cm) after T1 under D1 conditions, while the shortest ones were estimated as (33.99 and 27.88 cm) after T3 under D1 conditions. SH/R trait ranged from 4.22 to 6.85 after T2 and T3 applications under D2 conditions, respectively. Moreover, RL trait ranged from 5.97 to 12.48 cm after T3 and T1 under D2 and D1 conditions, respectively. Also, RW and R. no. had the maximum values of 0.21 g and 24.33 g after T1 and T3 applications under D1 conditions, respectively, while the minimum ones were scored at 0.08 g and 9.67g after T2 under D0 conditions.

Based on flag leaf traits, seven parameters (L no., FLL, FLW, FLL/FLW, FLA, FLP, and FLAn) were detected. The FLL, FLA, and FLP parameters recorded the highest values (15.22 cm, 17.30 cm^2^, and 32.97 cm) after T1 application under D0 conditions, while the lowest records were found at 7.15 cm, 3.92 cm^2^, and 15.40 cm after T3 under D0 conditions, respectively. Also, after application by T1 under D2 conditions, FLL/FLW gave the maximum records (20.55), and the best FLAn was recorded at 55.94° after application with T3 application under D2 conditions. Based on spike and spikelets traits, S. no. and FS parameters had the highest values (~43.00 and ~32.00) after application by T1 under D0 and D2, respectively.

Multivariate analysis including partial contribution plot, biplot, and T^2^ contribution plots showing Hotelling’s T^2^ values were assessed as illustrated in Figure 9. Figure 9a shows the partial correlation plot among yield traits of the 12 treated *T. aestivum* samples under different water regimes. The highest contribution of yield traits such as (FS, S no. FLL, and FLW) were in *T. aestivum* samples under D2 condition and FWR, FLL/FLW, SH/R, FLAn, L no. and WCR parameters were recorded to have the highest contribution under D0 condition. FWSH, FLW, SS, R no., WCSH, RW, FLA, and PL under D1 condition. Figure 9b represents the biplot of 22 yield traits in which the first component PC1 was scored as 42.3% and the second component PC2 as 19.9% of the total variation. Under D2 condition, the most significant vectors were WCSH, DWR, DWSH, FLAn, SH/R, FWR, FS, S.no., and R. no. Strong positive correlations were found among FS & S. no., WCSH & DWSH & DWR and FLAn & SH/R parameters, while the negative correlations were recorded between WCSH with SS. Under D1 condition, the most important vectors were SHL, FLW, RW, FLL/FLW, RL and FL. RW showed a strong correlation with FLW and FLL/FLW, while SS recorded a negative interconnection with R. no., FS, and S. no.

PCA mix-based Hotelling’s T^2^ multivariate control charts were assessed the proportion of yield traits of 12 treated *T. aestivum* samples under different water regimes as illustrated in Figure 9c,d. The highest parameter contribution was L no. (~0.35) in sample 8 after T3 application under D1 condition and FLL/FLW in after T0 application under D2 condition. SH/R and FLAn traits showed moderate contribution in sample 12 after T3 application under D2 condition. RL, FWR and, WCR parameters were contributed to sample 7 after T2 application under D1 condition. Figure 9d showed Hotelling’s T^2^ values of yield trait which most of them found around the median except samples 10 and 12 after T1 and T3 application under severe drought condition (D2) which increased over the upper control limit (UCL) ~ 7.9 at α = 0.05.

#### 2.5.2. Kernel or Grain Attributes and Their Contribution to Drought Stress Tolerance

Ten kernel or grain parameters of 102 DAS of *T. aestivum* under different water regimes with loading coefficients and partial contributions plots. GW, GP, GA, W, Circ., and Round parameters recorded the highest values after T2 application under D1 and D2 conditions as recorded in Table 5. The loading coefficient plot, represented in Figure 10a, showed that W, GW, and GA had strong positive correlation among them; also, Round is positively correlated with Circ. which had a negative correlation with AR. In addition, GP, Feret, and L showed strong positive associations with each other but had a moderate positive correlation with 20-Gwt. The partial contribution of GW, W, GA, Round, AR, and Circ. were recorded mostly under D2 conditions, while GP, Feret, L, and 20-Gwt showed their contributions under D1, as expressed in Figure 10b.

#### 2.5.3. Drought Tolerance Indices and Their Intercorrelations with and Contributions to Drought Stress

Eleven drought tolerance traits of 102 DAS of *T. aestivum* under different two drought stresses (D1 and D2), respectively, were estimated in Table 6. MP, YSI, RDI, DRI, and YI drought indices recorded the maximum values at 22, 3.69, 2.87, 5.12 and 1.41, while the TOL, SSI, and YRR indices had the minimum values of −20.00, −7.69 and −2.69, respectively, after application of T1 under D1 condition. The scatter plot with heatmap and partial contributions plot assessed the intercorrelation among drought tolerance indices as shown in Figure 11a,b.

The scatter plot matrix showed the density ellipses in each individual scatter plot and the red circles contain about 95% of the data as represented in Figure 11a. The green color assumed the positive correlation, the purple color assumed the negative correlation, while the size of the circles indicated the significance. Highly positive correlation was detected among MP and GMP and STI, TOL and SSI, STL and HARM, YSI and RDI and DRI and YI and YRR and TOL and SSI. On the contrary, TOL was negatively correlated to all the drought tolerance indices, except SSI and YRR. The partial contribution plot showed strong contribution of some indices such as YI, TOL, DRI, RDI, SSI, YRR, YSI, and MP under D2 condition. On the other hand, STI, GMP, and HARM were the most effective under D1 condition, as observed in Figure 11a,b.

### 2.6. Bioinformatic Analysis

#### 2.6.1. Function Assessment of the qRT-PCR-Associated Genes

In order to anticipate the biological functions of the five drought responsive genes, the sequence of qRT-PCR primers was searched against the genome sequence of *T. aestivum.* Subsequently, other databases like Ensembl Plants, Phytozome, NCBI, InterPro, and KEGG generated further functional annotations for these genes. In the given context, it can be shown that these *TaZFP34*, *TaWRKY1*, *TaMAPK3*, *TaLEA*, and *TaP5CS1* genes are related with Basic-leucine zipper domain, Transcriptional regulator ATRX homolog, Mitogen-activated protein kinase 3, Late embryogenesis abundant protein and delta l-pyrroline-5-carboxylate synthetase genes, respectively. The first gene *TaZFP34* is characterized by the presence of two domains CCAAT/enhancer-binding protein C/EBP (InterPro: IPR031106), and Basic-leucine zipper domain (InterPro: IPR004827). Moreover, the *TaWRKY1* gene is recognized by five domains P-loop containing nucleoside triphosphate hydrolase (InterPro: IPR027417), helicase superfamily 1/2, ATP-binding domain (InterPro: IPR014001), ADD domain (InterPro: IPR025766), Helicase, C-terminal (InterPro: IPR001650) and SNF2-related, N-terminal domain (InterPro: IPR000330). Furthermore, the *TaMAPK3* genes are marked by four domains protein kinase domain (InterPro: IPR000719), mitogen-activated protein (MAP) kinase, JNK (InterPro: IPR008351), tyrosine-protein kinase, catalytic domain (InterPro: IPR020635) and protein kinase-like domain (InterPro: IPR011009). In addition, the *TaLEA* gene is characterized by the presence of next three domains immunoglobulin-like fold (InterPro: IPR013783), water stress and hypersensitive response domain (InterPro: IPR013990) and late embryogenesis abundant protein, LEA-14 (InterPro: IPR004864). Besides, the *TaP5CS1* gene is distinguished by eight domains GPR domain (InterPro:IPR000965), glutamate/acetylglutamate kinase (InterPro:IPR001057), delta l-pyrroline-5-carboxylate synthetase (InterPro: IPR005766), aldehyde/histidinol dehydrogenase (InterPro: IPR016161), aldehyde dehydrogenase domain (InterPro: IPR015590), aspartate/glutamate/uridylate kinase (InterPro:IPR001048), aldehyde dehydrogenase N-terminal domain (InterPro: IPR016162) and glutamate 5-kinase/delta-1-pyrroline-5-carboxylate synthase (InterPro: IPR005715).

#### 2.6.2. Putative Tissue Expression Patterns of Drought-Responsive Genes in *T. aestivum*

To understand the potential roles of the five qRT-PCR primers that associated wheat genes in different tissues, their expression patterns were evaluated based on the transcriptome expression database in *T. aestivum* transcripts, as shown in Figure 12 and Appendix A. The results showed that the gene *TaZFP34-TraesCS1D02G312200* was favorably expressed across many wheat tissues, particularly in Awns—Ear emergence followed by First leaf sheath—Tillering stage, First leaf blade—Seedling stage, Spikelets—50 percent spike, Awn—50 percent spike, Roots—Flag leaf stage and Peduncle—Ear emergence. Moreover, the *TaWRKY1-TraesCS1B02G161400* was highly expression in Roots—Flag leaf stage then Awn—50 percent spike, Shoot axis—Milk grain stage, Spike, Internode #2—Milk grain stage, Stigma and Ovary, Internode #2—50 percent spike, Radicle—Seedling stage and Roots—50 percent spike. Furthermore, the *TaMAPK3-TraesCS4A02G106400* was highly expression in Glumes, Flag leaf sheath—50 percent spike, First leaf sheath—Tillering stage, Flag leaf sheath—Ear emergence, Internode #2—Milk grain stage, Fifth leaf blade (senescence)—Milk grain stage and Fifth leaf blade—Ear emergence. In addition, the putative expression of *TaLEA-TraesCS3A02G150800* gene was highly in Embryo proper, Grain—Hard dough, Grain—Ripening stage, Endosperm, Radicle—Seedling stage and First leaf sheath—Tillering stage. At the end, the *TaP5CS1-TraesCS3B02G395900* gene was highly expression in Anther followed by Shoot apical meristem—Seedling stage, Fifth leaf sheath—Fifth leaf stage, Peduncle, Awns—Milk grain stage and fifth leaf blade night (+0.25 h) 22:15, as shown in Figure 12 and Appendix A.

## 3. Discussion

Plant development and overall productivity are negatively impacted by drought-induced stress, which poses a serious threat to agricultural productivity. One environmentally friendly way to increase wheat production during drought conditions is to cultivate resistant wheat varieties. Via the modification of morphological, physiological, biochemical, and molecular traits, the Trichoderma species have demonstrated extraordinary potential for promoting plant development, enhancing systemic resistance, and reducing the negative effects of drought stress on plants [47]. Due to the induction of genes by Trichoderma spp., the priming response is dependent on the genetic control of the plant. Four different time stages can be used to categorize plants’ responses to Trichoderma. The induction of distinct plant genes results in a distinct cellular response at each stage [48].

The study performed by Abdenaceur, et al. [49] demonstrated that the strain of *T. harzianum* (OL587563) exhibited the highest level of performance in terms of phytohormone synthesis and hydrolytic enzymes. This finding implies that the strain may be a major factor in the improvement of plant development as a potential biofertilizer. In the current study, wheat seedlings treated with Trichoderma showed improved resistance to water stress. When plants were treated with *T. harzianum*, their deep roots became more numerous and enhanced the uptake of nutrients.

*Trichoderma* can produce phytohormones, including indole-3-acetic acid (IAA) and gibberellic acid (GA) which help maintain the balance of the plant phytohormone network and, consequently, affect the signaling pathways. These findings corroborate those of previous researchers [50,51], who discovered that certain Trichoderma strains generate gibberellin and IAA, a plant growth hormone. Bach, et al. [52] demonstrated the critical significance that microorganisms’ fixation of nitrogen plays in promoting plant growth. In the same line, it has been shown that *T. harzianum* had the ability to fix nitrogen, our result is supported also by Vaccaro, et al. [53]. However, the strain of this study was unable to solubilize phosphate, and agronomists have also become interested in phosphate-solubilizing microbes, which have been employed as a soil inoculum to enhance plant growth [54]. In the same manner, the Trichoderma isolate acts as a biocontrol agent by suppressing infections and also had the capacity to create extracellular enzymes like amylase, cellulase, and proteases (parasitism), which break down organic materials and absorb nutrients from plant tissues and competing with other organisms for a particular niche (nutrients, plant tissues, etc.). Furthermore, it facilitates the entry of fungal hyphae [51]. An aspartic protease was discovered to be strongly expressed in *T. harzianum* when the cell walls of *Pythium ultimum* and *Botrytis cinerea* were present, according to a proteomic research [55]. Our results were in the line with the study of Rahman, et al. [56], who studied the extracellular enzymes of two strains of *T. harzianum* MC2 and *T. harzianum* NBG, which were able to produce amylase and cellulase enzymes.

The goal of the reaction solution’s reflexing process in the synthesis of nanoparticles is to heat the reaction for an extended period without losing solution volume. Since the sample has completely melted, which is thermodynamically desirable because it speeds up the reaction, controls the crystals, and yields nanoparticles [57]. In the current work, rice husk, an agricultural waste product, was used to create roughly 96% of the amorphous pure silicon oxide powder using a green synthesis method. This study offers a straightforward, low-cost approach for producing high-purity amorphous nano-silica using reflexing and three hours of calcination at 700 °C. Characterization using FT-IR, SEM, XRD, EDX, and UV methods revealed the nanoscale crystal size. Sodium, oxygen, and silica content can be determined by using EDX spectroscopy, a technique that also provides information about the purity of the pattern and the percentage weight of the elements present in silica oxide powder made from rice husk ash. According to the EDX technique’s results, the percentage weight of impurities is 3%, and the silica element has a high percentage weight of 21.09%, indicating the appearance of a high-intensity peak. These findings demonstrated that rice husk ash contained highly pure silica oxide. These outcomes were in line with Ali and Drea [57]. The morphology of the sample was regulated, as demonstrated by SEM micrographs. The average particle size of the spherical silica oxide nanoparticles is less than 100 nm. Our findings concurred with the X-ray diffraction results of A Ajeel, et al. [58], which demonstrated the sample powder’s nanoscale particle size.

SiO-Si (siloxane group) at wavelengths 2114, 1637, and 1046, and symmetric vibrations at 793, and 618 cm^−1^ was shown to be responsible for the bending vibration of Si-O in this study based on FT-IR spectroscopic analysis of nano-silica particles. These findings accorded with those of Imoisili, et al. [59], whose FT-IR spectra of the nano-silica powder revealed the asymmetric vibration of SiO-Si (the siloxane group), the symmetric vibration at 808.20 cm^−1^ caused by the bond Si-O-Si, and the 484.14 cm^−1^ attributable to the bending vibration of Si-O. The number of electric charges on a nanoparticle’s surface is also counted using the zeta potential. The study found that the silica nanoparticles’ zeta potential was negatively charged (−26.3 mV) and stable.

Water deficiency can affect a variety of wheat morphological characteristics, including root characteristics like length, density, fresh and dry weight, and cuticle tolerance, as well as leaf characteristics like area, shape, size, pubescence, expansion, senescence, waxiness, and cuticle tolerance [17]. All the morphological features in the current study demonstrated a noticeable reduction during the period of severe drought (D2). Alternatively, following T1 application, there was a small improvement in leaf attributes and an increase in plant height, shoot and root traits (FWSH, DWSH, R. no., RW, FWR, and DWR). Biostimulants have been shown by Gul, et al. [60] to increase root biomass in conditions of stress. The promotion of nutritional intake and metabolic activity by the biostimulant is responsible for this increase.

Drought stress induces a range of adverse physiological and metabolic responses in plants [61]. Plants have evolved a spectrum of adaptive strategies to endure stress conditions. These adaptations encompass alterations in metabolic activities, structural modifications of cellular membranes, specific gene expression, and the synthesis of secondary metabolites [62]. Plant water status is a critical determinant and regulator of overall plant performance. It is well documented that plant growth is hindered under water stress due to altered plant-water relations [63]. Crucially, RWC is strongly associated with both the plant’s physiological activities and the soil water status, making it an essential trait for evaluating drought tolerance across various plant cultivars [64]. When plants encounter a drought, Trichoderma can increase their RWC, which helps them stay hydrated and reduce the impacts of water shortage [65]. The study found that the water content (WCSH and WCR) maximized in T0 under D0 conditions, started to decline in D1, then maximized in D2 conditions. Compared to other treatments, the water relations of the wheat samples’ shoot, and root improved following the application of T1 under D2.

It has been demonstrated that a plant’s levels of chlorophyll can be used as an indication to determine its drought resistance [66]. As the severity of the drought increased, T2 and T3 treatments had a discernible impact on the pigmentation (Chl a, Card, and TP) as an adaptive strategy. Drought considerably decreased all four wheat cultivars’ total pigment, Chl a + b, Chl a/b, Chl b, and Chl a during heading in both seasons, as demonstrated by Ghanem and Al-Farouk [15]. Furthermore, compared to the single application of PGPR or SiNPs, Alharbi, et al. [67] examined the combined application of (PGPR at 950 g ha^−1^; *Pseudomonas koreensis* MG209738 and *Azotobacter chroococcum* SARS 10) and silica nanoparticles (SiNPs) at 500 mg L^−1^ to mitigate the adverse effects of saline water on barley plants grown in salt-affected soil. The physiological characteristics of barley plants, such as their relative chlorophyll content (SPAD) and relative water content (RWC) were improved by the simultaneous application of SiNPs and PGPR.

Under stress conditions, changes in permeability to both electrolytes and non-electrolytes can indicate significant alterations in the plasma membrane’s protein and lipid components [68]. This adaptive response plays an important function in the plant’s ability to cope with environmental stresses. Drought stress affected membrane electrolyte leakage (EL) in all *T. aestivum* samples, and it showed the minimal values after T1 application under both drought stresses (D1 and D2).

To counteract the potential damage from ROS during periods of limited water availability, plants employ a range of antioxidant defense mechanisms, which include both enzymatic and non-enzymatic systems. Non-enzymatic antioxidants such as water-soluble reductants (e.g., proline, phenolics, glutathione, and ascorbate) and lipid-soluble antioxidants (e.g., carotenoids and tocopherols) play a key role in this defense. Additionally, enzymatic antioxidants such as catalase (CAT), polyphenol oxidase (PPO), and peroxidase (POD) are crucial for scavenging excess ROS, thus preventing oxidative damage [69]. In the current study, T2 and T3 applications to drought-stressed wheat plants significantly improved osmolytes and antioxidant contents compared to plants just treated with drought stress. For *T. harzianum*-treated wheat plants, the activation of stress-related signals such as TFC, TPC, and DPPH accumulation, as well as the CAT enzyme, which improved the plant hydration level. Biplot multivariate analysis confirmed the intercorrelations among morpho-physiological and biochemical traits. Strong positive correlations were found between Card and Chla, ProL and CAT, and EL and TFC, while the negative correlations were recorded among Card and Chl a, ProL and CAT with LA and LW and EL and TFC under drought stress.

Responses to drought stress vary greatly depending on the genetic background. Trichoderma has been reported to be essential in controlling the amounts of abscisic acid (ABA) during drought stress, which helps plants tolerate more water stress. One significant plant hormone that regulates the transpiration process in plants is ABA [70]. Previous research in wheat under drought stress revealed the presence of two ABA-responsive genes: the late embryogenesis abundance gene, *TaLEA* and the ABA-signaling negative regulator gene, *TaPP2C6* [71]. Additionally, P5CS, the pro biosynthesis gene, was suggested to be an ABA-responsive gene in Arabidopsis [72]. Enzymes implicated in osmolyte production pathways, such as delta-1-pyrroline-5-carboxylate synthetase (*P5CS*), the main enzyme in the proline metabolic pathway, can have their activity modulated by Trichoderma [73]. Proline content (ProL) and *TaP5CS1* expression in this study peaked following the administration of T1 and T3 under D1 conditions as well as T0, T2, and T3 under D2 conditions. Furthermore, Itam, et al. [74] found that *TaLEA*, *TaPP2C6*, and *TaP5CS* expression levels significantly increased under drought-stress conditions, beginning with DT4, suggesting that these genes were activated in response to drought-induced accumulation of ABA.

Unexpectedly, most of the wheat’s drought-upregulated ZFP genes are mostly expressed in the roots [75]. An essential organ for a plant’s adaptability to drought stress is its root. The primary roots can continue to elongate, albeit more slowly, under extreme dehydration stress when shoot growth is substantially or completely hindered [76]. As a result, the ratio of roots to shoots increases, and the plant’s water balance is maintained. In wheat, it has also been shown that under moderate and severe dehydration stress, the ratio of root-to-shoot length increases by more than 50% [77]. The molecular players that control this differential response are still unknown, though. Primary root elongation is known to be sustained by an increase in abscisic acid (ABA) concentration at low water potentials [76,78].

In this investigation, *TaZFP34* was upregulated in treated wheat samples under moderate drought stress after T3 followed T1 then T2 applications. As a result, the root length (RL) showed the highest length and elongation in the same trend of treatment applications. The overexpression of the gene stimulates drought tolerance machinery. This is based on findings showing increased expression of some drought-upregulated transcription factors positively affects root development [79]. Our results came in agreement with them. Furthermore, Chang, et al. [80] examined the role of *Triticum aestivum* ZFPs (*TaZFP34*, *TaZFP22*, and *TaZFP46*) that are mostly expressed in roots and that are induced by abiotic stress. Stresses such as oxidative damage, dehydration, excessive salinity, and cold increased the expression of *TaZFP34* in roots. A higher root-to-shoot ratio was the outcome of overexpressing *TaZFP34* in wheat roots, a condition that is seen when plants adjust to drying soil. Significant changes were seen in the expression of several genes that could be involved in controlling root growth in the roots of *TaZFP34* overexpressing lines. Specifically, there was a considerable decrease in the transcript levels of *TaRR12D*, *TaRR12B*, and *TaSHY2*, which are homologues of established root growth negative regulators.

Numerous biological processes in plants, including abiotic and biotic stress responses, have been linked to the WRKY transcription factor family. There are several physiological and biochemical processes in plants where the WRKY transcription factors have been elaborated. In wheat samples under D1 conditions, T2 application demonstrated the highest *TaWRKY1* expression in our investigation. To limit oxidative damage under drought stress, *TaWRKY1* expression also boosted total pigments, Chl a, and some enzyme activities including CAT and POD. It also decreased the cell membrane, which serves as an osmoprotectant, to lessen the impacts of osmotic stress. Similar findings were made by Yu, et al. [81], who found that under drought stress, *TaWRKY1-2* silencing in wheat raises the MDA content, lowers proline and chlorophyll contents and antioxidant enzyme activity, and inhibits the expression levels of antioxidant (*TaPOD*, *TaCAT*, and *TaSOD* (Fe)) and stress-related genes (*TaP5CS*). Furthermore, Yu and Zhang [82] demonstrated that *TaWRKY46* affects wheat’s (*Triticum aestivum* L.) ability to withstand abiotic stress. Different abiotic stimuli and hormone treatments, such as cold (4 °C), abscisic acid (100 μM ABA), PEG-induced stress (20% polyethylene glycol 6000), hydrogen peroxide (10 mM H_2_O_2_) and, salt (100 mM NaCl), differentially affected the transcription levels of the *TaWRKY46* gene.

*TaMPK3* overexpression significantly decreases the growth-inhibiting effects of ABA by decreasing drought tolerance and wheat sensitivity to ABA. Overexpression lines exhibit reduced survival rates, shoot fresh weight, and proline content under drought stress; however, at the seedling stage and at the adult stage, they exhibit decreased grain width and 1000 grain weight in both glasshouse and field conditions. Drought tolerance is increased by *TaMPK3*-RNAi [83]. Our results were in line with other research showing a drop in FWSH and ProL content following T2 treatment in D1 conditions.

Environmental stimuli like low temperature, salt stress, drought, and ABA can cause the expression of late embryogenesis abundant (*LEA*) proteins, which are widely expressed in plant roots, stems, leaves, and other tissues, as well as seeds, during late embryonic development. Quantitative real-time PCR (qRT-PCR) was used to identify the expression patterns of 16 *TaLEA* genes in eight subfamilies in order to gain insight into the roles of these genes in growth, development, and stress responses [84].

In this investigation, *TaLEA* genes were overexpressed and may cause an elevation in both WCSH and WCR content after T3 application under D1 condition. Our results in agreement with Silletti, et al. [85] concluded that the *LEA2* gene is a key regulator of drought stress responses and was highly expressed in *T. harzianum* T22 inoculated Durum wheat under severe water stress. Moreover, Wang, et al. [86] concluded that *TaLEA* genes improve the stress resistance of wheat by maintaining cellular osmotic pressure and protecting cell membranes and other biological macromolecules from damage. Furthermore, *TaLEA-1A*, a *LEA* gene that is variably expressed in imbibed seeds of the strong-dormancy variation Hongmangchun 21 (HMC21) and, the weak-dormancy variety Jing 411 (J411) was discovered by Lei, et al. [87]. *TaLEA-1A* is substantially expressed in wheat seeds. They concluded that *TaLEA-1A* might control germination and seed dormancy by modulating the balance between gibberellic acid and abscisic acid, based on their analysis of endogenous hormone levels and expression patterns.

In the context, BAR database was used to predict the tissue expression patterns of these previous genes were favorably expressed across many wheat tissues. Also, we used other databases such as InterPro, phytozome and KEGG pathway to predict and determine the putative domains in these previous gene sequencings. Therefore, we found these genes have been potentially linked to a variety of vital biological functions, such as the response to environmental stressors, proline accumulation, plant growth, development, and defence response, as in several other species such as Arabidopsis, cotton, poplar, rice, hazelnuts [88,89,90,91,92,93]. Also, our results were in agreement with Abdelhameed, et al. [94] who found that the genes *TraesCS7D02G326300*, *TraesCS7B02G230100*, and *TraesCS7A02G329600* were expressed differently in various wheat tissues, with the spikelets (50 percent spike), stigma, and ovary showing the highest expression.

Any factor that influences the metabolism of a plant at any point during its growth and development could influence its yield, which is the result of the combined effects of its metabolic responses. The primary selection criteria for drought tolerance are grain yield, as it is the most crucial financial aspect for wheat plants [95]. In this investigation, the agronomic attributes of 22 yield traits of 102 DAS of *T. aestivum* under different water regimes were recorded. After T1 application (*T. harzianum*), the root and shoot traits such as fresh weight and water content and other parameters (FWSH, FWR, DWSH, WCSH, WCR, RL, and RW) were recorded as the best records under normal and severe drought (D0 and D2) circumstances. These findings were consistent with those of Sallam, et al. [96] and Ahmad, et al. [97], who reported that water stress reduced wheat yield and yield-related features. Typically, wheat flag leaves are thought to be the primary organ of photosynthesis. Thus, degree of succulence, specific area, and leaf area index are crucial adaptive markers of stress tolerance [15]. In drought-stressed cereal crops like barley, wheat, sorghum, and oats, genotypes with a narrow leaf size and an erect leaf angle have been linked to increased photosynthesis [98]. *T. harzianum* enhanced the flag leaf characteristics in the current study in both normal and severe drought (D0 and D2) conditions, including length, area, perimeter, and flag leaf length-to-breadth ratio. Large T^2^ readings would suggest important variations brought on by drought stress.

Larger grains are found in tetra- and hexaploid wheat species, and Goriewa-Duba, et al. [99] revealed the morphological differences between wheat kernels from different genotypes. A variety of models were put out to measure seed form, considering differences in roundness and circularity in addition to a visual analysis of kernel shape. The two models’ results show that wheat kernels can be categorized into two groups: “rounded” kernels and “nearly elongated” kernels [100]. In this study, kernel or grain parameters GW, GP, GA, W, Circ. and Round parameters recorded the highest values after T2 application under D1 and D2 conditions. Kernels of all treated wheat plants had a higher AR more than (1.8) except kernels of T2 treated plants, thus all kernels were nearly elongated. With T3 application of wheat plants under different water regimes, the lower Circ. and Round values and the elongation of kernels increased.

A score of less than one for the stress-sensitive index (SSI) indicated high cultivar tolerance to stress [101]. Under drought conditions, high YSI index values suggested possible yield [102]. According to Ladoui, et al. [102], high TOL values suggested potential yield in non-stressed circumstances. The MP, YSI, RDI, DRI, and YI drought indices in this study showed the highest records following the application of T1 under D1 conditions, whereas the TOL, SSI, and YRR indices showed the lowest records. Our findings were in agreement with those of Ghanem and Al-Farouk [15], who found that the SSI, MP, STI, and YSI all showed that Giza 171 > Misr 1 > Misr 3. They concluded that Sakha 95 was the most tolerant wheat cultivar and Misr 3 was the least.

## 4. Materials and Methods

### 4.1. Experimental Materials

*Trichoderma harzianum* was acquired from the Mycology lab, Botany Department, Faculty of Science, Mansoura University, Egypt. Additionally, cultural and morphological identifications to confirm the strain. All micromorphological data were examined on cultures grown on potato dextrose agar PDA for five days at 28 °C. The microscopic examination of conidiophores and conidia was made from slide preparations. Rice husk was from the locally agricultural market for the biological synthesis of silica nanoparticles. Spring wheat cultivar Summit (*Triticum aestivum* L.) seeds were brought from the Department of Plant Production, College of Food and Agriculture Sciences, King Saud University, Riyadh, Saudi Arabia.

### 4.2. Screening of Trichoderma Drought Tolerance

Using the method outlined by Aujla and Paulitz [103], polyethylene glycol (PEG) 6000 MW was dissolved in a potato dextrose agar (PDA) medium to induce desiccation conditions. By mixing 0, 15, 20, 25, 30, and 35 g of PEG with 100 mL of PDA, various matric water potentials were generated. Three replicates of the experiment were used for Trichoderma, and the growth findings were indicated on a scale of ++++ (totally bushy), + to +++ (growing hyphal growth), and − (no growth) [51].

### 4.3. Testing the Effectiveness of T. harzianum as a Biostimulant: Plant Growth-Promoting Trait Characterization

#### Determination of Phytohormone-like Compounds

Using L-tryptophan as a precursor, the Bric, et al. [104] method was used to screen Trichoderma isolate for their effective capacity to generate indole acetic acid (IAA). In potato dextrose broth medium (PDB) enriched with 1 g L^−1^ L-tryptophan, Trichoderma is inoculated. After being incubated at 28 °C, the culture was shaken at 150 rpm. Using a colorimetric technique, production was qualitatively assessed following 3 days of incubation. The synthesis of IAA was discerned by the appearance of a pink or red color, and the absorbance was determined at 530 nm using a spectrophotometer (JSR-100C, Gongju-City, Republic of Korea).

Gibberellic acid (GA) calorimetric determination was performed using a standard technique [105,106]. Gibberellin content was calculated by establishing a standard curve and using gibberellic acid as the reference, and the absorbance was measured at 254 nm.

### 4.4. Determination of Nitrogen Fixation and Ammonia Production

A medium lack of nitrogen (pH 7.2) was used to assess nitrogen-fixing capabilities. A six-millimeter disc from a pure culture of Trichoderma was injected in the middle of a Petri dish that contained media for fixing nitrogen. The test for nitrogen fixing abilities was deemed positive after three days of inoculation if the colony could develop regularly on the selective medium [107].

The production of ammonia in peptone water was tested using the Bakker and Schippers [108] approach. A broth culture was inoculated and then incubated for 72 h at 28 °C with 10 mL of peptone water added. Nessler’s reagent (1 mL) was thereafter added to each tube. The tint changed from yellow to a brownish orange color, signifying a positive result for ammonia production.

### 4.5. Determination of Phosphate Solubilization Potential

The process of phosphate solubilization was qualitatively screened using the solid medium NBRIP agar containing tricalcium phosphate as an insoluble inorganic phosphorus source, as per the methodology described by Nautiyal [109]. The 6 mm agar disc used in the assay was cut from a 5-day-old fungal culture of each strain and inoculated into the medium. The disc was then incubated at 28 ± 2 °C. Positive solubilization of mineral phosphate was shown by a clear zone surrounding the fungal colony [110].

### 4.6. Determination of Extracellular Enzymatic Activity

The amylase, cellulase, and protease activities of the isolate were assessed in solid culture media. A medium for amylase detection was used to measure the production of amylase [111]. The plates were flooded with 5 mL of iodine solution for 15 min. after being incubated at 28 °C for 72 h. Amylase production was measured by observing the development of a pale-yellow halo surrounding colonies. A cellulose agar medium was used to quantify cellulase by streaking on medium and incubated at 28 °C for a period of 2–5 days [112]. Following that, the plates were inundated with 0.2% aqueous Congo red and destained for 15 min. with 1 M NaCl in order to assess the activity, as per [113]. Casein hydrolysis in milk agar was used to identify protease producers [114]. The presence of a transparent halo surrounding the colonies served as an indicator of enzyme activity because the milk agar is opaque.

### 4.7. Eco-Friendly Rice Husk Ash Waste Silica Nanoparticle Synthesis: Preparation of Silica Oxide

According to the method described by Jyoti, et al. [115], through the procedures of co-precipitation and leaching, amorphous nano-silica was produced using rice husk ash (RHA). After being removed from a rice mill, the rice husk was burned for six hours at 700 °C in a muffle furnace. The obtained RHA was treated with 6 N HCl for 2 h and agitated in a magnetic stirrer during the leaching process. Filter paper and deionized water were then used to filter the leached RHA. The filtration process was finished when the pH of RHA reached 7. The neutralized RHA was treated with 2.5 N NaOH solutions at 80 °C for two hours while being stirred with a magnetic stirrer to obtain sodium silicate. Amorphous nano-silica formed when concentrated H_2_SO_4_ was added to the resultant sodium silicate to adjust the pH to 2. Subsequently, the nano-silica was annealed at 100 °C for 24 h. Amorphous nano-silica (SiO_2_NPs) could eventually be obtained.

### 4.8. Optical, Structural, and Morphological Characterizations of Nano-Silica

#### 4.8.1. UV–Visible Spectroscopy Analysis and Zeta Potential

Using a UV-vis spectrophotometer (ATI Unicam 5625 UV/VIS Vision Software V3.20, Tokyo, Japan), the synthesis of SiO_2_NPs was scanned in the 200–900 nm range. Using Zeta Potential Analyzer (Zetasizer ver. 7.01, Malvern Instruments, Westborough, MA, USA) a dynamic light scattering device, the zeta potential was determined.

#### 4.8.2. SEM-EDX Analysis: Scanning Electron Microscopy (SEM) and Energy-Dispersive X-ray (EDX)

Scanning electron microscopy (SEM JEOL JSM-6510, Tokyo, Japan) with EDX (Oxford X-Max 20, Wycombe, UK) was used to analyze the particle size, shape, and morphology of the synthesized SiO_2_NPs.

#### 4.8.3. Fourier Transform Infrared (FTIR) Spectroscopy, X-ray Diffraction (XRD), and Thermogravimetric Analysis (TGA) of Silica Nanoparticles

FTIR measurements in the 400–4000 cm^−1^ range were conducted at room temperature at the Spectrum Unit of the Faculty of Science, Mansoura University, Egypt, using a computerized recording Fourier transform infrared (FTIR) spectrometer (ThermoFisher Nicolete IS10, Waltham, MA, USA). A tube operating at 30 kV was utilized to get X-ray diffraction (XRD) scans with a Bragg’s angle (2 h) ranging from 5 to 80 degrees using a Malvern Panalytical X’Pert PRO system (Malvern, Worcestershire, UK) with Cu Ka radiation (k = 1.540 A°). Thermogravimetric analysis (TGA) data were collected in a nitrogen atmosphere between 25 and 800 °C, using a TA Instrument 2050 thermogravimetric analyzer (New Castle, DE, USA) set to heat at a rate of 10 °C/min.

### 4.9. Field Experiment

#### 4.9.1. Experimentation: Experimental Sit and Climate

A pot study was executed in a greenhouse located at the Botany Department, Faculty of Science, Mansoura University, Egypt, from December 2023 to April 2024. The temperature was between 18 and −30 °C with a photoperiod of 12:12 h and a relative humidity of 60–75%. The soil used for pot filling with the following properties: pH = 8.59, electric conductivity (EC) = 323.84 ppm, water potential (WP) = 5.1%. Soluble anions such as chloride (CL^−^) = 0.52 meq/100 g, biocarbonate (HCO_3_^−^) = 0.41 meq/100 g and soluble cations including: magnesium (Mg^++^) = 0.14 meq/100 g, calcium (Ca^++^) = 0.24 meq/100 g, sodium (Na^+^) = 0.7 meq/100 g and potassium (K^+^) = 0.09 meq/100 g were measured.

#### 4.9.2. Experimental Design and Treatments

A field experiment was conducted to examine the impacts of grain bio-priming with *T. harzianum* spore suspension 3 × 10^−8^ and SiO_2_NPs (600 ppm) alone or combined in a mixture, as shown in Figure 13. Before being inoculated with treatments, seeds were first surface sterilized with a 5% sodium hypochlorite solution and then washed with autoclaved distilled water [116]. Seeds were sown under uniform soil moisture, and bio-nanopriming by silica nanoparticles and *T. harzianum* spore suspension was applied for 1 h before sowing. Initially, 15 seeds were seeded in each pot at identical distances. After emergence, ten seedlings in each pot were retained to maintain a uniform stand. The experimental design was completely randomized into four treatments in three replicates, ten plants per pot: T0 (control: distilled water), T1 (*T. harzianum*), T2 (SiO_2_NPs (600 ppm)), and T3 (*T. harzianum* + SiO_2_NPs (600 ppm)). Three irrigation regimes (D0, no drought (100% field capacity (FC)), D1, mild drought (50% FC), D2, severe drought (30% FC)) were used in this experiment to investigate how treatments affected the traits-related water stress (WS) tolerance of wheat plants 65 DAS (days after sowing).

#### 4.9.3. Crop Husbandry

Seeds were sown in plastic pots filled with 10 kg of soil composed of sand and clay at a ratio at ratio of 2:1, respectively. The temperature was between 18 and 30 °C with a photoperiod of 12:12 h and a relative humidity of 60–75%. A dose of NPK = (72.79, 23.81, and 378.61 mg/kg soil, respectively) was dispensed against the proposed dose for wheat at the time of pot filling. Fertilizers in the form of 5% calcium and phosphorus, 0.57% magnesium and 1% sulfur, the full dose was applied at sowing. Water losses were compensated by delivering water to maintain the required FC. To guarantee proper moisture levels, the control group of wheat plants received irrigation throughout the experiment, whereas the drought group received supplemental irrigation. From the seedling stage to the heading stage of wheat in soil culture, a 42-day drought treatment was applied. Fifty days after sowing, the same treatment was sprayed on the leaves once. In order to investigate various morphological, biochemical, physiological, and molecular parameters, data was collected 65 days after sowing at the heading stage and 102 DAS at the harvest stage.

### 4.10. Morphological and Physiological Parameters of the Heading Stage

Data on the root and shoot lengths of randomly chosen plants were immediately recorded following the sampling of 65-day-old plants process. The root and shoot’s fresh weight was determined using an electric balance. Using the same electric balance, the dry biomass of oven-dried samples at 80 °C until constant weight was observed as well as the water content of the shoot and root. The morphological traits of the wheat plants in both normal and stressful conditions were determined, such as the following: PL (plant length; SHL (shoot length); RL (root length); SH/R (shoot/root length ratio); L. no. (leaf number); LL (leaf length); LW (leaf width); LA (leaf area); LP (leaf perimeter); RW (root width); R. no (root number); FWSH (fresh weight of shoot); DWSH (dry weight of shoot); FWR (fresh weight of root); DWR (dry weight of root); WCSH (water content of shoot), and WCR (water content of root). Image J 1.50i program was used for length, area and perimeter measurements.

Physiological, biochemical, and metabolic traits of the treated wheat samples were calculated, such as the following: EL (electrolyte leakage); Chl a (chlorophyll a); Chl b (chlorophyll b); Card (carotenoids); TP (total pigments); Carb (carbohydrates); ProT (protein); ProL (proline); TFC (total flavonoids content); TPC (total phenolic content); CAT (catalase); POD (peroxidase), and PPO (polyphenol oxidase).

Electrolyte Leakage (EL)

In accordance with the methodology outlined by Bajji, et al. [117], electrolyte leakage was assessed. The following formula was used to determine the electrolyte leakage:Electrolyte leakage (%) = (EC2) − (EC1)/(EC3) × 100

### 4.11. Biochemical Analyses

#### 4.11.1. Determination of Total Proteins and Total Carbohydrates

Using a tissue grinder, 0.2 g of fresh leaves were ground in 5 mL of cooled 50 mM Tris HCl (pH 7.8) that was immersed in an ice bath to extract protein. The total protein content was determined using Bradford [118] methodology, whereas the total carbohydrate content of the dried leaves was determined using the Hedge, et al. [119] method.

#### 4.11.2. Determination of Pigments and Total Proline

The amount of chlorophyll and carotenoids was measured according to Barnes, et al. [120] and Wellburn [121]. Using a spectrophotometer with both blank and extract, absorbance was measured at 470, 646, and 665 nm. For proline, using 1 mL of glacial acetic acid, and 1 mL of ninhydrin reagent, the absorbance was measured at 510 nm. The standard curve was created using proline equivalents [122].

#### 4.11.3. Determination of Antioxidant Enzymatic Activities

Polyphenol oxidase (PPO), peroxidase (POD), and catalase (CAT) are three antioxidant enzymatic activities related to ROS scavenging that were analyzed as previously described [116,123]. The concentration of these antioxidant enzyme activities was determined using wheat leaves that were detached from 65-day-old plants and homogenized in 50 mM potassium phosphate buffer (pH 7.8). The units of measurement for PPO, POD, and CAT activities were Unit per gram of fresh weight.

#### 4.11.4. Determination of Non-Enzymatic Antioxidants

Methanolic extracts were used to measure the amounts of non-enzymatic antioxidants in leaves [124]. The Folin–Ciocalteu method was used to measure the total phenolic content spectrophotometrically at 765 nm. The results were reported as gallic acid equivalents (GAEs), which are represented as mg gallic acid/g of dry weight [125]. The aluminum chloride method was used to quantify the total flavonoid content at 510 nm. Concentrations were reported as mg (QE)/g of dry leaf weight and estimated as quercetin (QE) equivalents [126]. The assay for DPPH radical scavenging capacity was conducted in accordance with Rosidah, et al. [127]. The inhibition ratio, which was determined using the following equation:DPPH radical scavenging activity (%) = [(Abs control − Abs sample)/Abs control] × 100

### 4.12. Molecular Analysis: Expression Analysis Using Quantitative RT-PCR

RNA extraction: In liquid nitrogen, frozen leaf samples were ground into a fine powder. Following the manufacturer’s instructions, Plant RNA Reagent (Invitrogen, Carlsbad, CA, USA) was used to isolate total RNA from the samples. In accordance with the manufacturer’s instructions (Toyobo, Osaka, Japan), 500 ng of total RNA were reverse transcribed using ReverTra Ace qPCR RT Master Mix with gDNA remover to create first-strand cDNA.

qRT-PCR for genes responsive to drought: Quantitative real-time PCR (qRT-PCR) was performed using the QuantiTect SYBR Green PCR kit (Qiagen, Hilden, Germany) methodology to determine the expression levels of the related genes in the samples. Target transcript accumulation with respect to the housekeeping gene can be measured via SYBR green, which binds to double-stranded DNA and causes the amplification products to glow.

Primer design: Using the Phytozome database (https://www.phytozome.net, accessed on 10 February 2024), sequence homologs of five drought-responsive genes were compared to hexaploid wheat sequences from the International Wheat Sequencing Consortium. The genes are as follows: (1) *TaZFP34*, which regulates root growth [80]; (2) *TaWRKY1*, which transduces ABA signaling [128]; (3) *TaMPK3*, Mitogen-activated protein kinase, *MPK3* [129]; (4) *TaLEA*, a protein abundant in late embryogenesis [74]; and (5) *TaP5CS1*, a Proline biosynthetic gene.

For the purpose of determining the relative transcript levels of the genes of interest, the wheat housekeeping gene *TaActin* was chosen as an internal reference gene [72]. The sequences that exhibited the highest degree of similarity with the genome of hexaploid wheat were chosen. Gene-specific primer sets are shown in Table 7, and primer sequences were created using NCBI PrimerBlast (https://www.ncbi.nlm.nih.gov/tools/primer-blast/, accessed on 15 February 2024).

The PCR program included 40 cycles of 95 °C for 15 s, 59 °C for 30 s, and 72 °C for 30 s after a starting temperature of 98 °C for 2 min. By raising the temperature from 72 to 95 °C at a rate of 0.05 °C s^−1^, a melting curve was created. The study employed four biological replicates, and the 2^−ΔΔCt^ method [130] was employed to assess the relative expression levels of five genes that are responsive to drought. *TaActin* was utilized as an internal standard for normalization.

### 4.13. Yield-Related Agronomic Traits of Wheat

#### 4.13.1. Yield Attributes

At maturity, the crop was harvested at 102-days-old, and data regarding root and shoot length, leaf, and seed parameters were recorded, along with the number of spikelets per spike, fertile spikelets and sterile spikelets, with at least three biological replicates per pot for both the control group and two drought groups. Shoot length (SHL), root length (RL), plant length (PL), shoot-to-root length ratio (SH/R), root width (RW), and root number (R. no.) were among the various yield characteristics that were directly measured in order to evaluate the plants’ yield attributes. Shoot and root fresh and dry weight, and then shoot and root water content was calculated. Furthermore, a number of flag leaf characteristics were identified, such as L. no. (leaf number); FLL (flag leaf length); FLW (leaf width); FLL/FLW (flag leaf length-to-weight ratio); FLA (leaf area); FLP (flag leaf perimeter); FLAn (flag leaf angle).

#### 4.13.2. Grain Phenotypic Traits

The grain or kernel morphology was investigated using a seed tester after the seeds were fully dried and threshed in an air-drying room. Grain measurements were taken in all wheat treatments to measure the following characteristics: 20-GWt (20-grains weight); GW (grain width); GP (grain perimeter); GA (grain area); L (length of grain major axis); W (length of seed minor axis); AR (aspect ratio); Circ. (circulatory); Round (roundness) and; Feret (Feret diameter).

#### 4.13.3. Drought Tolerance Indices

The algorithms were utilized to create several drought response indices according to these characteristics. After determining of grain yield in g plant^−1^ under well-watered (Yp) and drought (Ys) conditions was determined, then the mean grain yield under well-watered (Ȳp) and drought (Ȳs) conditions could be calculated. Using these features, the different drought response indices were calculated, such as the following: mean productivity (MP); geometric productivity (GMP); stress susceptibility index (SSI); yield stability index (YSI); tolerance index (TOL); harmonic mean of yield (HARM); stress tolerance index (STI); drought resistance index (DRI); relative drought index (RDI); yield index (YI), and yield reduction ratio (YRR).

### 4.14. Bioinformatics Approaches

#### 4.14.1. Function Predictions of Wheat Genes Associated with Drought Responses

The primers for our candidate genes sequences were utilized to query the *T. aestivum* genomes we got from the NCBI website database (https://www.ncbi.nlm.nih.gov/genome/11; accessed on 25 August 2024). Then, the alignment sequence was compared using the available data from multiple NCBI GenBank, Phytozome, and Ensembl Plants databases to determine wheat qRT-PCR primers candidate genes. Phytozome v13 and Ensembl Plants were used to derive annotations for these genes’ probable roles [131,132]. NCBI blast sequences were produced against the genomes of different wheat species: *T. aestivum*, *T. turgidum*, *T. dicoccoides*, and *T. urartu*.

#### 4.14.2. Potential Tissue Expression Patterns of the Target Genes

Target genes (*TaZFP34-TraesCS1D02G312200, TaWRKY1-TraesCS1B02G161400, TaMAPK3-TraesCS4A02G106400, TaLEA-TraesCS3A02G150800* and *TaP5CS1-TraesCS3B02G395900*) that are linked with drought responses in wheat were investigated for their potential tissue-specific expression patterns. Their expressions were analyzed from the *T. aestivum* transcript expression database, that includes data of seventy one tissues and organs, including First leaf sheath—Tillering stage, Internode #2—Milk grain stage, Shoot apical meristem—Seedling stage, Grain—Milk grain stage, First leaf blade—Seedling stage, Flag leaf blade—Full boot, Awn—50 percent spike, flag leaf blade night (−0.25 h) 06:45, Shoot axis—Flag leaf stage, Fifth leaf blade—Flag leaf stage, Third leaf sheath—Three leaf stage, Internode #2—Ear emergence, Anther, Spike, Coleoptile, Stigma and Ovaryl, Roots—Flag leaf stage, Fifth leaf sheath—Flag leaf stage, Root apical meristem—Three leaf stage, Flag leaf sheath—Ear emergence, Roots—Three leaf stage, Axillary roots—Three leaf stage, Flag leaf sheath—50 percent spike, Radicle—Seedling stage, Roots—50 percent spike, Third leaf blade—Three leaf stage, Spikelets—50 percent spike, Root apical meristem—Tillering stage, Grain—Ripening stage, Awns—Ear emergence, Glumes, Glumes—Ear emergence, Leaf ligule, Flag leaf blade—50 percent spike, Internode #2—50 percent spike, Fifth leaf sheath—Fifth leaf stage, Fifth leaf blade night (−0.25 h) 21:45, Grain—Soft dough, Flag leaf blade (senescence)—Dough stage, Flag leaf blade night (−0.25 h) 06:45—Flag leaf stage, Flag leaf blade (senescence)—Ripening stage, First leaf blade—Tillering stage, Shoot apical meristem—Tillering stage, Shoot axis—First leaf stage, Roots—Seedling stage, Shoot axis—Milk grain stage, Fifth leaf blade—Fifth leaf stage, Flag leaf blade—Ear emergence, Flag leaf blade night (+0.25 h) 07:15, Fifth leaf blade night (−0.25 h) 21:45, Shoot axis—Tillering stage, Stem axis—First leaf stage, Endosperm, Peduncle, Peduncle—50 percent spike, Peduncle—Ear emergence, Flag leaf sheath—Full boot, Flag leaf blade—Flag leaf stage, Lemma, Lemma—Ear emergence, Awns—Milk grain stage, Fifth leaf blade night (+0.25 h) 22:15, Flag leaf blade—Milk grain stage, Grain—Hard dough, Flag leaf sheath—Milk grain stage, Embryo proper, Fifth leaf blade (senescence)—Milk grain stage, Roots—Tillering stage, Shoot axis—Full boot, Fifth leaf blade—Ear emergence, First leaf sheath—Seedling stage and First leaf sheath—Seedling stage from distinct developmental phases. Wheat plant Electronic Fluorescent Pictograph Browsers were used to generate expression profiles (Wheat eFP browsers; https://bar.utoronto.ca/efp_wheat/cgi-bin/efpWeb.cgi; accessed on 1 September 2024) [94,131,132].

### 4.15. Statistical Analysis

A randomized complete block design with three replications was used to gather the phenological, physiological, biochemical, agronomic, and drought tolerance indices data of *T. aestivum* with four different applications under various water regimes. The general linear model was used to do a two-factor (treatment × water regimes) analysis of variance (ANOVA) on the data. After that, the mean differences were assessed using SPSS 16.0 (IBM, Armonk, NY, USA) and Duncan’s multiple range test at *p* ≤ 0.05 [133]. The mean ± standard deviation (SE) and lowercase Latin characters were used to indicate significant changes among different *T. aestivum* samples. Normality was assessed using Shapiro–Wilk normality testing at the 0.05 level. Figures of gene expression were plotted using GraphPad Prism 9 (GraphPad Software, Inc., San Diego, CA, USA). Cell plot, multivariate analyses, and intercorrelation including principal component analysis (PCA), score plot, biplot, partial contribution plot, loading coefficient plot, T^2^ contribution plots showing Hotelling’s T^2^ values with a significant level of α = 0.05 and scatter matrix plot with heatmap correlations of the studied data were visualized by JMP^®^, Version 17.2.0 (SAS Institute Inc., Cary, NC, USA, 2022–2023).

## 5. Conclusions

Our research suggested that the use of *T. harzianum* alone or in combination with SiO_2_NPs constituted a viable approach and is a useful tool for reducing the detrimental effects of drought on wheat productivity. By reducing soil salinity through a synergistic application, the physiological processes, as well as enzyme activities like POD and PPO, were stimulated, leading to an eventual increase in wheat production. Trichoderma and its host plants have a mutually beneficial connection that provides a wide range of strategies to reduce the effects of drought stress. These include boosting osmolyte synthesis, enhancing root growth, enhancing stomatal regulation, and modifying gene expression to increase stress tolerance in plants. Additionally, from mild-to-severe drought, the five responsive genes, *TaP5CS1*, *TaZFP34*, *TaWRKY1*, *TaMPK3*, and *TaLEA,* were consistently and quickly elevated, indicating that they are drought biomarkers. Using Electronic Fluorescent Pictograph Browsers, we matched these genes to wheat data from seventy-one tissues and organs, generating expression profiles for the wheat. We found that the examined genes have different domains involved in drought stress defence response, proline accumulation, plant growth, and development. It is also necessary to investigate the process by which Trichoderma colonizes plant roots to control signaling pathways that initiate defensive and developmental responses. Through improving our knowledge, we may make Trichoderma-based methods more widely used in sustainable farming practices, which will have a major positive impact on global farming communities’ welfare, conservation of the environment, and food security.

## Figures and Tables

**Figure 1 ijms-25-10954-f001:**
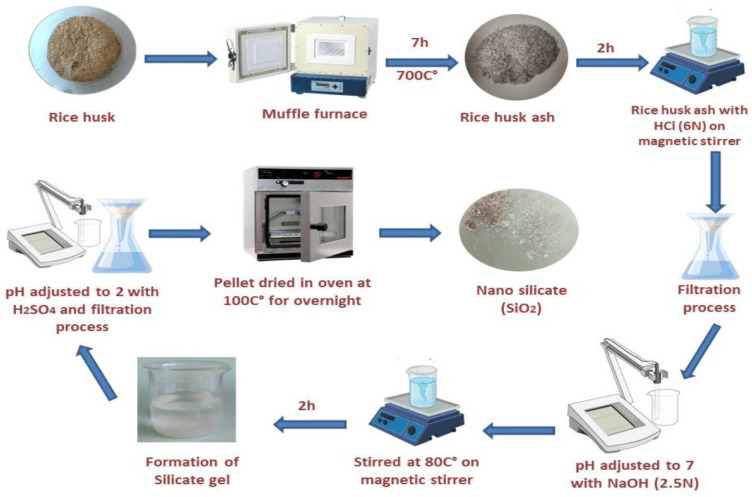
Stepwise of the green synthesis of silica nanoparticles (SiO_2_ NPs) from rice husk ash.

**Figure 2 ijms-25-10954-f002:**
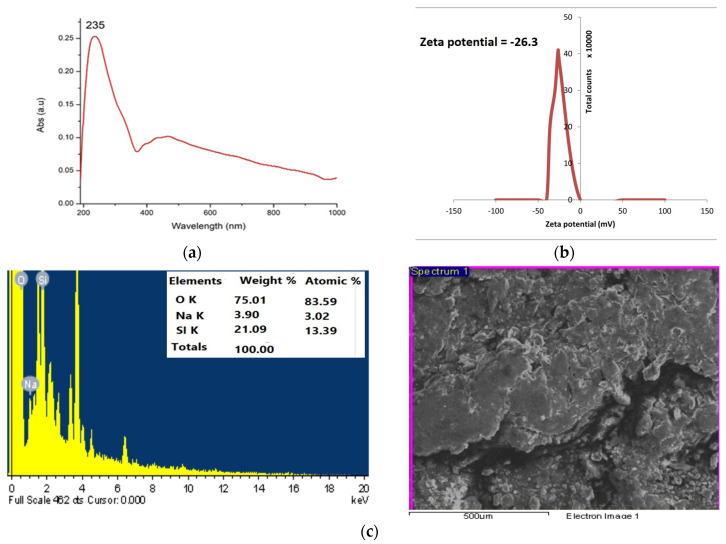
Optical and structural characterization of silica nanoparticles through: (**a**) ultra-violet (UV) spectroscopy; (**b**) zeta potential analysis; and (**c**) SEM-EDX analysis: scanning electron microscopy (SEM) and energy dispersive X-ray (EDX).

**Figure 3 ijms-25-10954-f003:**
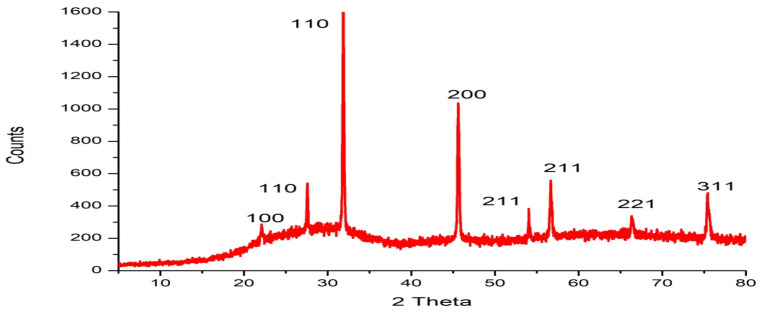
XRD analysis of silica nanoparticles.

**Figure 4 ijms-25-10954-f004:**
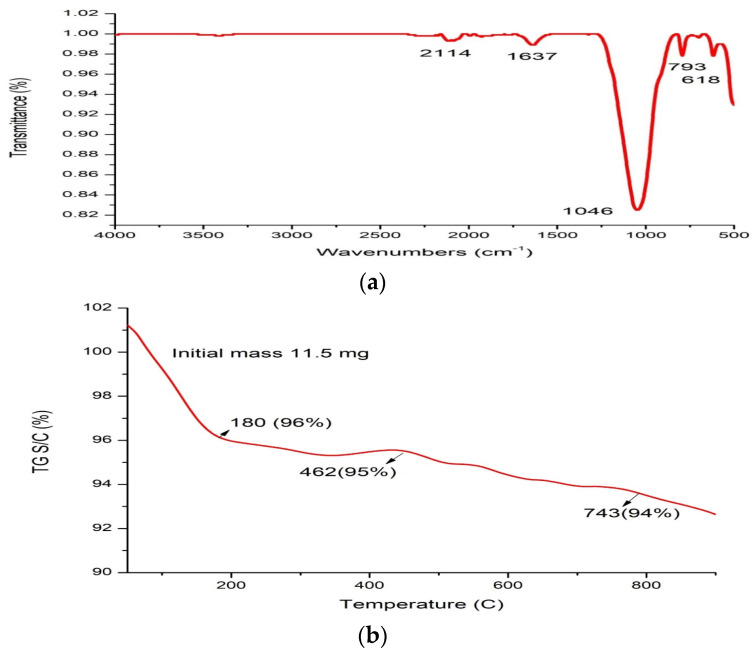
Chemical characterization of silica nanoparticles through (**a**) Fourier transform infrared (FTIR) spectroscopy; (**b**) thermogravimetric analysis (TGA).

**Figure 5 ijms-25-10954-f005:**
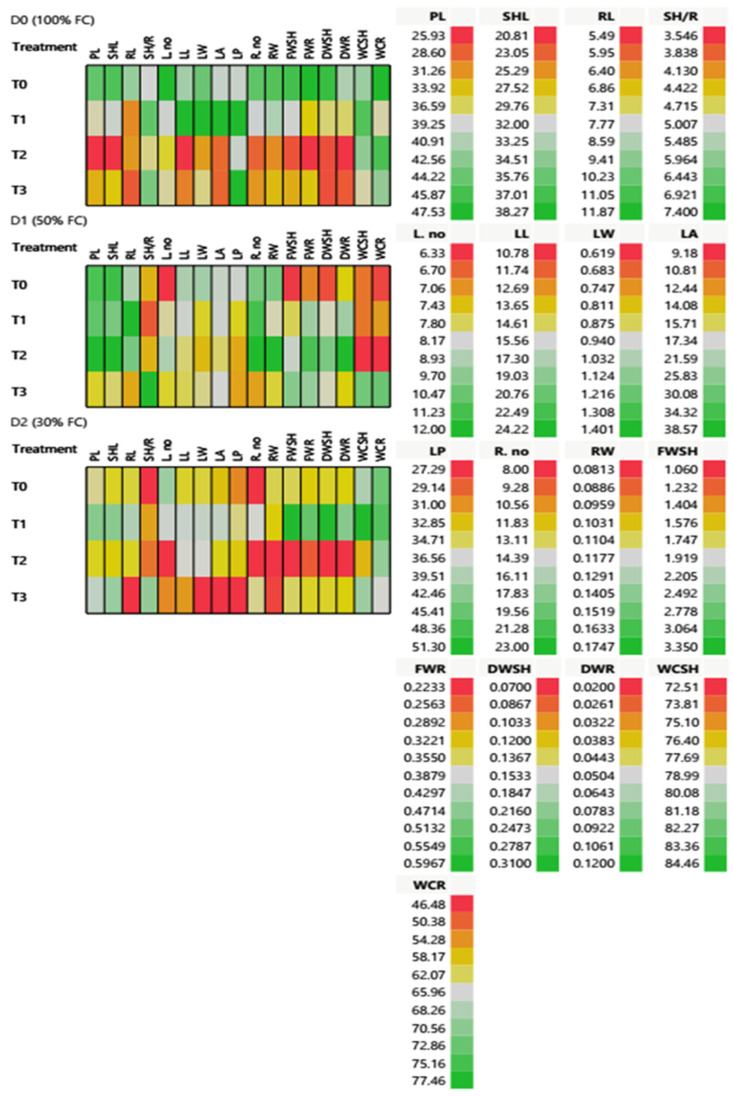
Cell plot of 18 morphological and phenotypic traits of 65 DAS of *T. aestivum* under different water regimes (D0, control; 100% FC; D1, mild drought with 50% FC; D2, Severe drought with 30% FC). T0 (control: distilled water), T1 (*T. harzianum*), T2 (SiO_2_NPs (600 ppm(), and T3 (*T. harzianum* + SiO_2_NPs (600 ppm)). Abbreviations: PL (plant length; SHL (shoot length); RL (root length); SH/R (shoot/root length ratio); L. no (leaves number); LL (leaf length); LW (leaf width); LA (leaf area); LP (leaf perimeter); RW (root width); R. no (root number); FWSH (fresh weight of shoot); DWSH (dry weight of shoot); FWR (fresh weight of root); DWR (dry weight of root); WCSH (water content of shoot), and WCR (water content of root). The dark red color indicates the highest values, while the dark green color shows the lowest ones.

**Figure 6 ijms-25-10954-f006:**
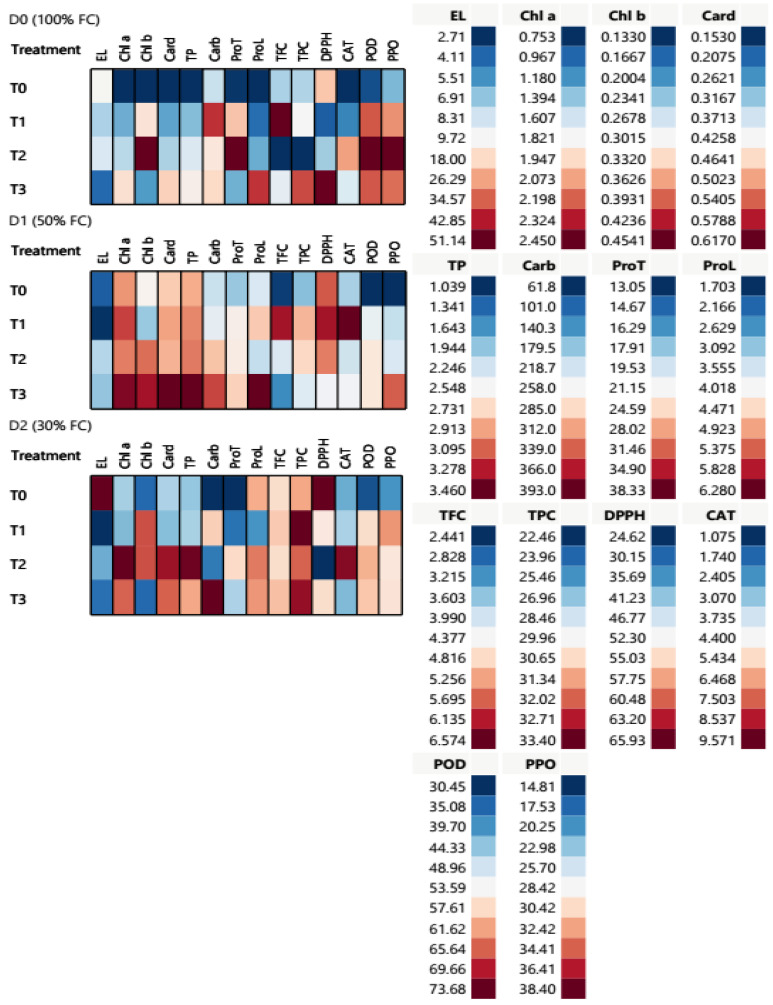
Cell plot of 14 physiological and biochemical traits of 65 DAS of *T. aestivum* under different water regimes (D0, control; 100% FC; D1, mild drought with 50% FC; D2, severe drought with 30% FC). T0 (control: distilled water), T1 (*T. harzianum*), T2 (SiO_2_NPs (600 ppm)), and T3 (*T. harzianum* + SiO_2_NPs (600 ppm)). Abbreviations: EL (electrolyte leakage); Chl a (chlorophyll a); Chl b (chlorophyll b); Card (carotenoids); TP (total pigments); Carb (carbohydrates); ProT (protein); ProL (proline); TFC (total flavonoids content); TPC (total phenolic content); CAT (catalase); POD (peroxidase), and PPO (polyphenol oxidase). The dark red color indicates the highest values, while the dark blue color shows the lowest ones.

**Figure 7 ijms-25-10954-f007:**
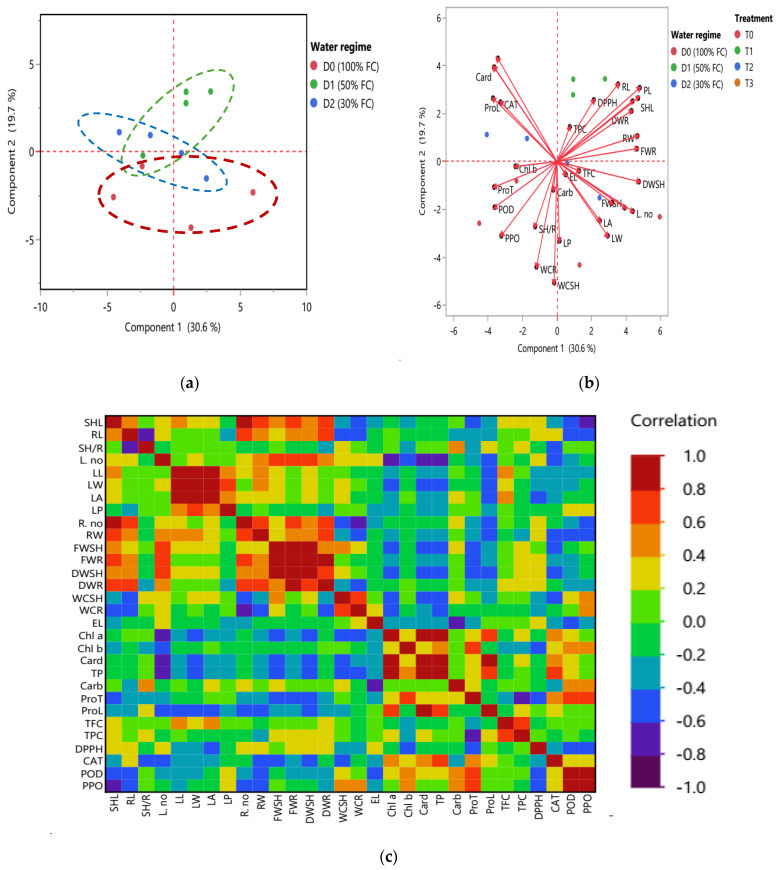
Principal component analysis (PCA): (**a**) score plot; (**b**) biplot of all 32 combined vegetative traits (morphological, physiological and biochemical traits; (**c**) heatmap of partial correlation among the most significant morpho-physiological parameters at the heading stage of 65 DAS of *T. aestivum* under different water regimes. The dashed blue, green, and red circles represent the distribution of the treated plants under different water regimes. The dots were *T. aestivum* under different water regimes of FC, and the vectors (red arrows) were parameters. The green-to-red color gradient indicated positive correlation, while the green-to-violet color gradient indicated the negative correlation (see scale at the above right corner). Abbreviations are provided in previous figures.

**Figure 8 ijms-25-10954-f008:**
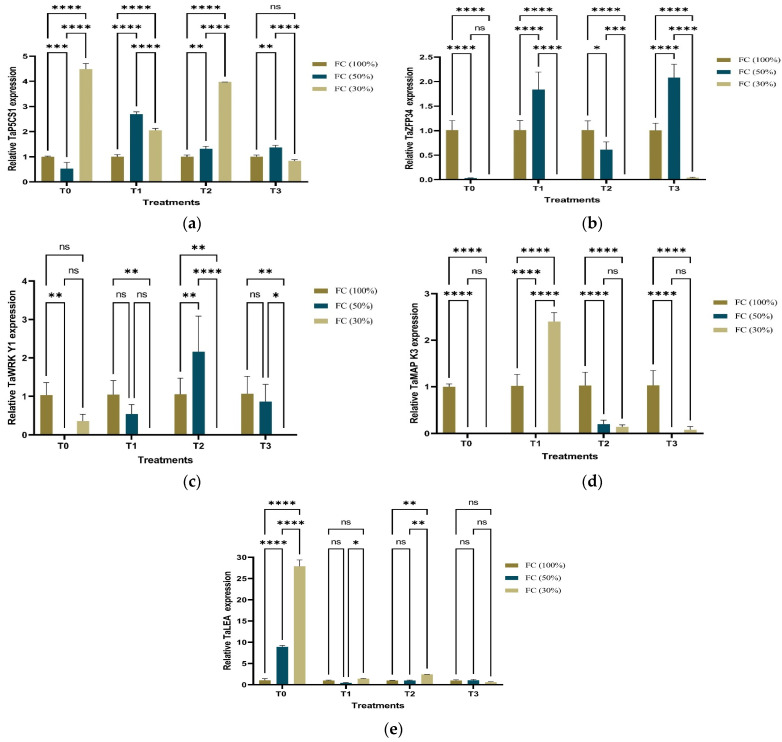
(**a**–**e**) Gene expression quantitative real-time PCR (qRT-PCR) of five drought-responsive genes at the heading stage of 65 DAS of *T. aestivum* under different water regimes. The genes are *TaP5CS1*, *TaZFP34*, *TaWRKY1*, *TaMPK3*, *TaLEA*, and the wheat house-keeping gene *TaActin*. ****, ***, **, * denote significant at *p* ≤ 0.0001, *p* ≤ 0.001, *p* ≤ 0.01, *p* ≤ 0.05, respectively, and ns donates non-significant difference.

**Figure 9 ijms-25-10954-f009:**
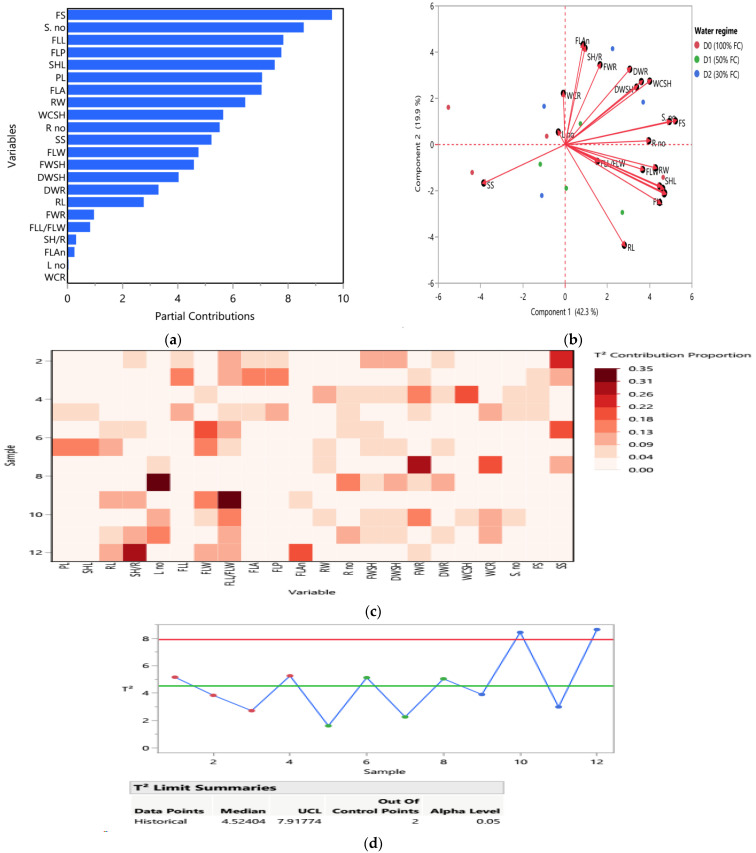
Agronomic attributes of 22 yield traits of 102 DAS of *T. aestivum* under different water regimes: (**a**) plot of partial contribution, (**b**) biplot, (**c**), and (**d**) T^2^ contribution plots showing Hotelling’s T^2^ values and the upper control limit (UCL) which represents in the red line while, the green line shows the median. Abbreviations: PL (plant length; SHL (shoot length); RL (root length); SH/R (shoot/root length ratio); L. no (leaf number); FLL (flag leaf length); FLW (leaf width); FLL/FLW (flag leaf length to weight ratio); FLA (leaf area); FLP (flag leaf perimeter); FLAn (flag leaf angle); RW (root width); R. no (root number); FWSH (fresh weight of shoot); DWSH (dry weight of shoot); FWR (fresh weight of root); DWR (dry weight of root); WCSH (water content of shoot) and WCR (water content of root); S. no (no. of spikelets/spike); FS (fertile spikelets) and SS (sterile spikelets).

**Figure 10 ijms-25-10954-f010:**
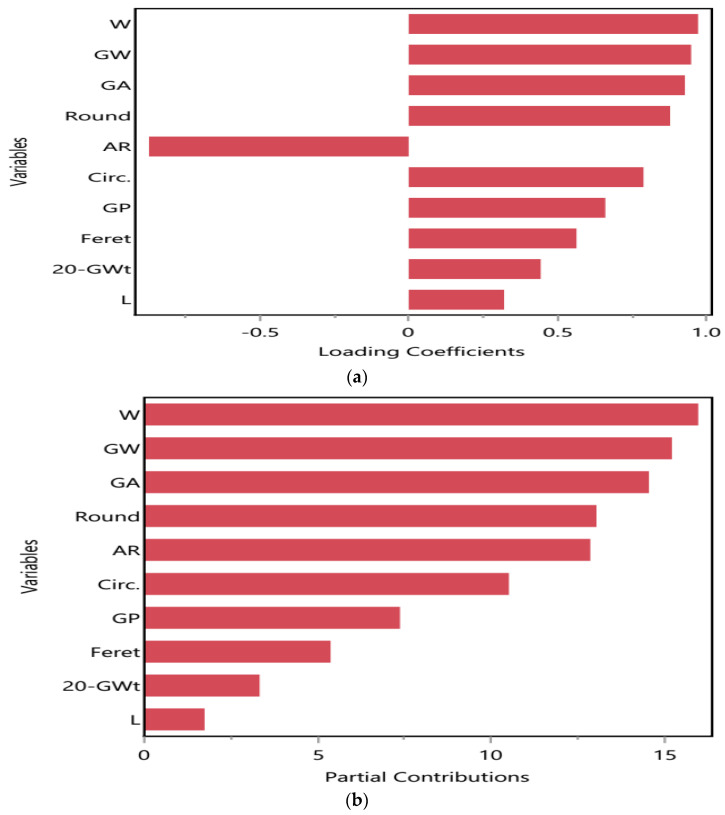
The kernel or grain parameters of 102 DAS of *T. aestivum* under different water regimes: (**a**) plot of loading coefficient and (**b**) partial contributions. Abbreviations: 20-GWt (20-grains weight); GW (grain width); GP (grain perimeter); GA (grain area); L (length of grain major axis); W (length of seed minor axis); AR (aspect ratio); Circ. (circulatory); Round (roundness) and Feret (Feret diameter).

**Figure 11 ijms-25-10954-f011:**
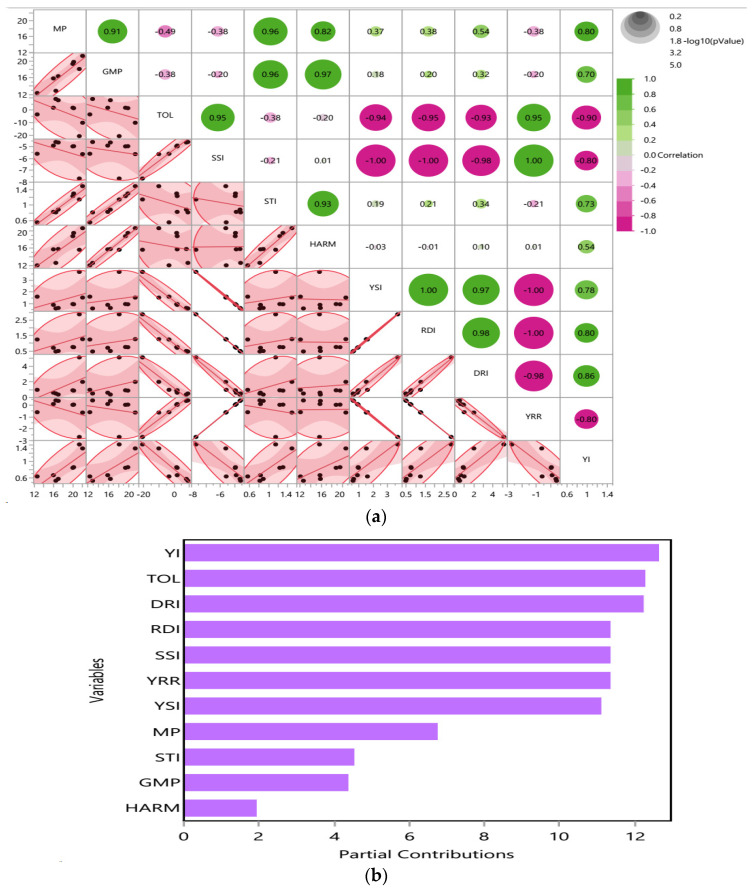
Drought tolerance traits of 102 DAS of *T. aestivum* under different water regimes: (**a**) scatter plot with heatmap and (**b**) partial contributions plot showing the intercorrelation between parameters. Abbreviations: mean productivity (MP); geometric productivity (GMP); tolerance index (TOL); stress susceptibility index (SSI); stress tolerance index (STI); harmonic mean of yield (HARM); yield stability index (YSI); relative drought index (RDI); drought resistance index (DRI); yield reduction ratio (YRR), and yield index (YI). The green color indicates positive correlation, purple color indicates the negative correlation, while the size of circles indicates the significance (see scale in the above–right corner).

**Figure 12 ijms-25-10954-f012:**
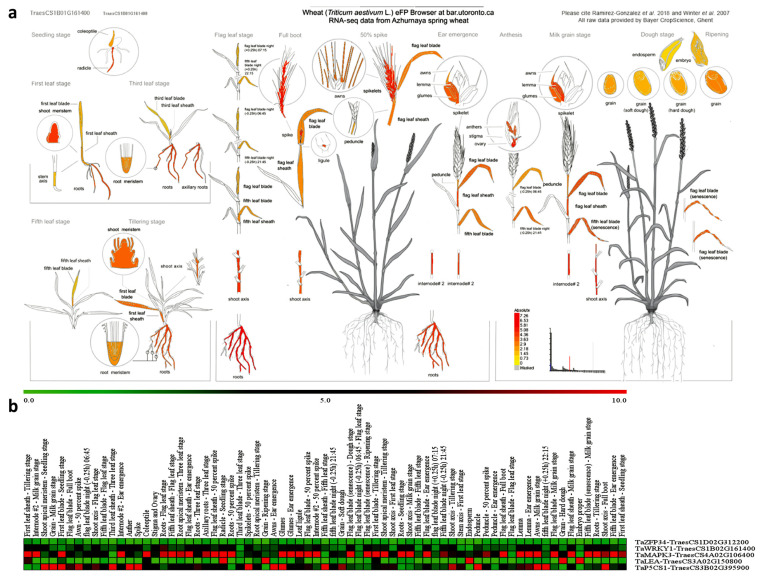
(**a**) The putative “wheat electronic fluorescent pictograph” tissue expression *of TaZFP34-TraesCS1D02G312200*, *TaWRKY1-TraesCS1B02G161400*, *TaMAPK3-TraesCS4A02G106400*, *TaLEA-TraesCS3A02G150800*, and *TaP5CS1-TraesCS3B02G395900* genes at different tissues and developmental stages; and (**b**) gene expression heatmap. The more intense the red color of the expression bar, the more gene expression detected according to [45,46].

**Figure 13 ijms-25-10954-f013:**
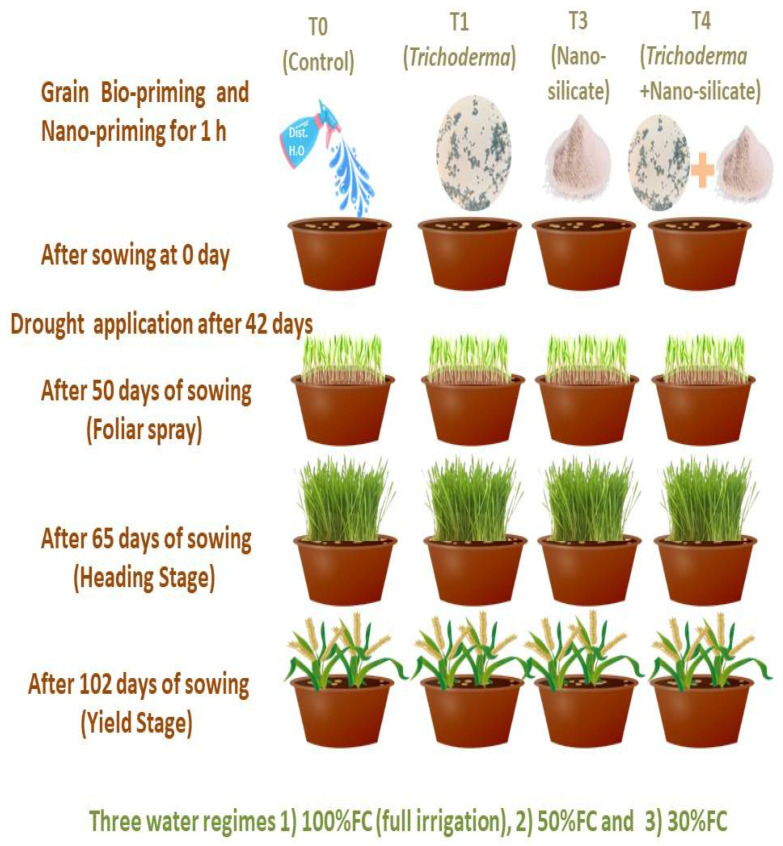
Experimental design and treatments of *T. aestivum* under different water regimes (D0, control; 100% FC; D1, mild drought with 50% FC; D2, Severe drought with 30% FC). T0 (control: distilled water), T1 (*T. harzianum*), T2 (SiO_2_NPs (600 ppm)), and T3 (*T. harzianum* + SiO_2_NPs (600 ppm)).

**Table 1 ijms-25-10954-t001:** Plant growth-promoting characterization of *Trichoderma harzianum*.

**Quantitative assay**	
Indole acetic acid (IAA) (µg/mL)	8.50 ± 0.59
Gibberellic acid (GA) (µg/mL)	28.62 ± 1.76
**Qualitative assay**	
Cellulase	+ve
Amylase	+ve
Protease	−ve
Phosphate solubilization	−ve
Ammonia	−ve
Nitrogen fixation	+ve

**Table 2 ijms-25-10954-t002:** Morphological and phenotypic traits of 65 DAS of *T. aestivum* under different water regimes at the heading stage.

Treatment		PL (cm)	SHL (cm)	RL (cm)	SH/R	L. No	Leaf Growth Parameter				R. No.	RW (cm)	FWSH (g)	FWR (g)	DWSH (g)	DWR (g)	WCSH (%)	WCR (%)
	Parameter						LL (cm)	LW (cm)	LA (cm^2^)	LP (cm)								
100% FC (D0)	T0	44.38 ± 2.05 ^a^	36.67 ± 1.53 ^a^	9.10 ± 1.88 ^a^	5.05 ± 0.29 ^a^	12.00 ± 2.00 ^a^	17.84 ± 1.20 ^b^	1.21 ± 0.12 ^b^	18.98 ± 2.67 ^b^	37.60 ± 2.59 ^b^	20.0 ± 1.53 ^a^	0.16 ± 0.03 ^a^	3.08 ± 0.14 ^a^	0.29 ± 0.06 ^a^	0.60 ± 0.05 ^a^	0.07 ± 0.01 ^a^	80.68 ± 0.85 ^c^	77.46 ± 1.22 ^a^
	T1	38.42 ± 1.12 ^b^	32.39 ± 0.98 ^b^	6.35 ± 1.65 ^a^	6.56 ± 2.28 ^a^	8.33 ± 0.88 ^a^	24.22 ± 0.42 ^a^	1.40 ± 0.09 ^a^	38.57 ± 1.48 ^a^	51.30 ± 1.57 ^a^	14.6 ± 2.91 ^b^	0.13 ± 0.02 ^a^	1.96 ± 0.11 ^b^	0.14 ± 0.03 ^b^	0.33 ± 0.02 ^b^	0.05 ± 0.00 ^b^	82.97 ± 0.77 ^a^	64.56 ± 9.68 ^a^
	T2	25.93 ± 2.21 ^c^	20.81 ± 1.23 ^d^	6.4 ± 1.67 ^a^	4.82 ± 1.45 ^a^	7.67 ± 1.76 ^a^	10.78 ± 0.62 ^c^	0.76 ± 0.06 ^d^	11.08 ± 0.56 ^c^	37.60 ± 0.89 ^c^	9.33 ± 1.76 ^d^	0.10 ± 0.01 ^a^	1.18 ± 0.14 ^d^	0.08 ± 0.02 ^b^	0.22 ± 0.03 ^b^	0.02 ± 0.01 ^c^	81.06 ± 0.68 ^b^	74.83 ± 1.92 ^a^
	T3	33.02 ± 3.41 ^d^	27.79 ± 2.85 ^c^	5.87 ± 2.23 ^a^	6.20 ± 1.73 ^a^	8.00 ± 0.00 ^a^	12.35 ± 0.37 ^c^	0.90 ± 0.19 ^c^	11.01 ± 1.21 ^c^	51.30 ± 0.76 ^c^	10.67 ± 0.88 ^c^	0.10 ± 0.01 ^a^	1.50 ± 0.25 ^c^	0.08 ± 0.01 ^b^	0.33 ± 0.06 ^b^	0.03 ± 0.00 ^c^	78.53 ± 0.92 ^d^	70.06 ± 1.02 ^a^
50% FC (D1)	T0	45.61 ± 2.21 ^a^	37.27 ± 1.91 ^a^	8.67 ± 1.25 ^b^	4.34 ± 0.04 ^a^	6.33 ± 0.88 ^a^	17.56 ± 2.60 ^a^	1.06 ± 0.04 ^a^	19.14 ± 1.84 ^a^	37.19 ± 5.21 ^a^	18.6 ± 1.20 ^a^	0.15 ± 0.04 ^a^	1.08 ± 0.19 ^d^	0.08 ± 0.00 ^b^	0.28 ± 0.11 ^d^	0.04 ± 0.01 ^b^	74.37 ± 0.82 ^b^	48.04 ± 9.03 ^b^
	T1	44.57 ± 2.45 ^a^	35.16 ± 2.31 ^a^	11.8 ± 1.24 ^a^	3.79 ± 0.41 ^a^	8.00 ± 0.58 ^a^	15.88 ± 5.00 ^a^	0.85 ± 0.20 ^a^	18.14 ± 7.92 ^a^	33.80 ± 9.69 ^a^	21.3 ± 6.36 ^a^	0.12 ± 0.00 ^a^	1.74 ± 0.19 ^c^	0.15 ± 0.02 ^a^	0.45 ± 0.33 ^a^	0.07 ± 0.02 ^b^	74.38 ± 3.19 ^b^	55.08 ± 7.60 ^b^
	T2	47.53 ± 2.78 ^a^	38.27 ± 1.69 ^a^	9.84 ± 1.76 ^b^	4.34 ± 0.70 ^a^	9.00 ± 1.53 ^a^	14.81 ± 2.22 ^a^	0.80 ± 0.07 ^a^	15.95 ± 3.93 ^a^	31.96 ± 5.04 ^a^	23.0 ± 6.03 ^a^	0.17 ± 0.01 ^a^	2.05 ± 0.29 ^b^	0.23 ± 0.04 ^a^	0.55 ± 0.30 ^c^	0.12 ± 0.02 ^a^	72.51 ± 4.11 ^b^	46.48 ± 2.74 ^b^
	T3	35.54 ± 2.33 ^b^	30.15 ± 3.40 ^a^	6.63 ± 1.16 ^c^	7.40 ± 2.92 ^a^	7.67 ± 1.67 ^a^	14.56 ± 1.86 ^a^	0.90 ± 0.14 ^a^	17.69 ± 3.85 ^a^	31.70 ± 3.93 ^a^	11.0 ± 0.58 ^a^	0.11 ± 0.03 ^a^	2.53 ± 0.22 ^a^	0.15 ± 0.03 ^a^	0.46 ± 0.22 ^b^	0.04 ± 0.01 ^b^	81.99 ± 0.54 ^a^	72.62 ± 3.48 ^a^
30% FC (D2)	T0	37.79 ± 0.26 ^b^	28.77 ± 1.42 ^b^	7.20 ± 1.26 ^b^	3.55 ± 0.70 ^c^	9.33 ± 1.20 ^a^	14.24 ± 1.10 ^b^	0.86 ± 0.05 ^a^	14.08 ± 3.65 ^b^	30.64 ± 3.26 ^b^	8.00 ± 2.01 ^a^	0.11 ± 0.00 ^a^	1.74 ± 0.38 ^b^	0.13 ± 0.04 ^b^	0.33 ± 0.08 ^b^	0.04 ± 0.01 ^b^	80.17 ± 4.73 ^b^	72.42 ± 5.73 ^a^
	T1	42.81 ± 0.82 ^a^	34.57 ± 1.08 ^a^	8.58 ± 0.49 ^a^	4.22 ± 0.35 ^b^	8.33 ± 1.45 ^a^	16.15 ± 0.76 ^a^	0.98 ± 0.20 ^a^	19.45 ± 1.92 ^a^	35.31 ± 2.13 ^a^	14.33 ± 2.67 ^a^	0.11 ± 0.01 ^a^	3.35 ± 0.38 ^a^	0.31 ± 0.05 ^a^	0.52 ± 0.12 ^b^	0.08 ± 0.02 ^a^	84.46 ± 1.39 ^a^	74.21 ± 3.14 ^a^
	T2	35.27 ± 0.51 ^c^	28.12 ± 0.21 ^b^	7.11 ± 0.85 ^b^	3.95 ± 0.31 ^c^	6.33 ± 0.33 ^a^	15.52 ± 1.10 ^b^	0.94 ± 0.04 ^a^	14.82 ± 1.16 ^a^	33.09 ± 3.31 ^b^	8.00 ± 2.63 ^a^	0.08 ± 0.02 ^a^	1.06 ± 0.12 ^b^	0.07 ± 0.01 ^b^	0.25 ± 1.66 ^a^	0.02 ± 0.00 ^b^	76.11 ± 2.61 ^b^	69.94 ± 13.16 ^a^
	T3	40.13 ± 1.04 ^b^	34.02 ± 0.74 ^a^	5.49 ± 0.99 ^c^	5.86 ± 0.72 ^a^	7.00 ± 0.58 ^a^	12.86 ± 0.56 ^c^	0.62 ± 0.08 ^a^	9.18 ± 0.78 ^c^	27.29 ± 1. 46 ^c^	13.67 ± 3.07 ^a^	0.08 ± 0.03 ^a^	1.76 ± 0.15 ^b^	0.13 ± 0.04 ^b^	0.34 ± 0.09 ^b^	0.04 ± 0.01 ^b^	80.63 ± 0.69 ^b^	65.83 ± 2.90 ^a^
**Two-Way ANOVA**																		
Interaction		****	***	ns	ns	ns	*	ns	**	*	*	ns	****	*	****	****	***	*
Treatment		***	**	ns	ns	ns	**	*	***	**	ns	ns	***	ns	ns	**	ns	ns
Water regimes		****	***	ns	**	**	ns	*	ns	ns	**	*	ns	****	ns	****	****	****

The results were recorded as mean of triplicates ± standard error (S.E). Different superscript letters refer to significant differences (*p* ≤ 0.05) (Duncan’s multiple range test), ****, ***, **, * denote significant at *p* ≤ 0.0001, *p* ≤ 0.001, *p* ≤ 0.01, *p* ≤ 0.05, respectively, and ns donates non-significant difference. (D0, control; 100% FC; D1, mild drought with 50% FC; D2, severe drought with 30% FC). PL (plant length), SHL (shoot length), RL (root length), SH/R ratio (shoot-to-root length ratio), L. no (leaf number), LL (leaf length), LW (leaf width), LA (leaf area), and LP (leaf perimeter), R. no (root number), RW (root width), FWSH (shoot fresh weight), FWR (root fresh weight), DWSH (shoot dry weight), DWR (root dry weight), WCSH (shoot water content) and), WCR (root water content).

**Table 3 ijms-25-10954-t003:** Physiological and biochemical traits of 65 DAS of *T. aestivum* under different water regimes at the heading stage.

Treatment		EL (%)	Chl a (mg/g)	Chl b (mg/g)	Carotenoids (mg/g)	Total Pigment (mg/g)	Carbohydrates (mg/g)	Protein (mg/g)	Proline (mg/g)	Flavonoids (mg/g)	Phenolic (mg/g)	DPPH (%)	CAT (U/g)	POD (U/g)	PPO (U/g)
	Parameter														
100% FC (D0)	T0	10.31 ± 1.20 ^a^	0.75 ± 0.12 ^c^	0.13 ± 0.01 ^d^	0.15 ± 0.03 ^d^	1.04 ± 0.16 ^d^	215.17 ± 6.21 ^c^	13.23 ± 0.38 ^d^	1.70 ± 0.04 ^d^	3.76 ± 0.06 ^b^	27.70 ± 0.79 ^c^	56.00 ± 0.71 ^b^	1.07 ± 0.03 ^d^	33.11 ± 0.96 ^c^	22.28 ± 0.64 ^c^
	T1	7.53 ± 0.68 ^b^	1.29 ± 0.02 ^b^	0.32 ± 0.05 ^b^	0.28 ± 0.02 ^c^	1.90 ± 0.04 ^c^	357.14 ± 12.35 ^a^	25.97 ± 1.63 ^b^	2.25 ± 0.04 ^c^	6.57 ± 0.14 ^a^	29.91 ± 0.50 ^d^	29.43 ± 3.45 ^d^	2.18 ± 0.08 ^c^	66.22 ± 1.15 ^b^	32.97 ± 0.99 ^b^
	T2	8.63 ± 1.26 ^b^	1.53 ± 0.44 ^b^	0.45 ± 0.06 ^a^	0.34 ± 0.07 ^b^	2.33 ± 0.56 ^b^	273.28 ± 10.00 ^b^	38.33 ± 1.47 ^a^	2.87 ± 0.07 ^b^	2.44 ± 0.09 ^c^	22.46 ± 0.98 ^b^	42.54 ± 2.72 ^c^	6.53 ± 1.11 ^a^	73.68 ± 2.86 ^a^	38.40 ± 1.03 ^a^
	T3	4.22 ± 1.97 ^c^	1.92 ± 0.29 ^a^	0.21 ± 0.02 ^c^	0.47 ± 0.06 ^a^	2.60 ± 0.34 ^a^	284.46 ± 10.31 ^b^	17.23 ± 0.22 ^c^	5.66 ± 0.50 ^a^	4.22 ± 0.53 ^b^	32.24 ± 0.62 ^a^	65.90 ± 0.85 ^a^	3.94 ± 0.11 ^b^	66.10 ± 2.81 ^b^	34.05 ± 1.86 ^b^
50% FC (D1)	T0	3.91 ± 0.90 ^b^	2.10 ± 0.15 ^a^	0.31 ± 0.07 ^a^	0.475 ± 0.02 ^a^	2.88 ± 0.24 ^a^	215.23 ± 10.08 ^d^	18.09 ± 0.59 ^c^	3.67 ± 0.15 ^b^	2.54 ± 0.12 ^b^	26.78 ± 0.41 ^c^	60.74 ± 1.27 ^a^	3.30 ± 0.11 ^d^	30.45 ± 0.88 ^b^	14.81 ± 0.43 ^c^
	T1	2.77 ± 0.53 ^c^	2.25 ± 0.17 ^a^	0.24 ± 0.02 ^a^	0.502 ± 0.04 ^a^	2.99 ± 0.22 ^a^	241.20 ± 5.20 ^c^	22.14 ± 0.53 ^b^	4.61 ± 0.81 ^b^	6.22 ± 0.58 ^a^	31.11 ± 0.58 ^a^	63.67 ± 1.61 ^a^	9.57 ± 0.33 ^a^	51.93 ± 1.72 ^a^	25.25 ± 0.87 ^b^
	T2	7.69 ± 0.72 ^a^	2.14 ± 0.28 ^a^	0.39 ± 0.02 ^a^	0.492 ± 0.08 ^a^	3.02 ± 0.36 ^a^	296.13 ± 28.35 ^b^	22.46 ± 0.81 ^b^	3.47 ± 0.06 ^b^	4.10 ± 0.32 ^b^	30.68 ± 0.43 ^a^	59.27 ± 2.04 ^a^	3.71 ± 0.30 ^c^	55.76 ± 1.61 ^a^	26.34 ± 0.98 ^b^
	T3	6.95 ± 1.78 ^b^	2.41 ± 0.43 ^a^	0.43 ± 0.02 ^a^	0.617 ± 0.12 ^a^	3.46 ± 0.54 ^a^	349.31 ± 25.61 ^a^	25.21 ± 1.19 ^a^	6.28 ± 0.38 ^a^	3.16 ± 0.67 ^b^	28.97 ± 0.52 ^b^	51.26 ± 1.45 ^b^	4.28 ± 0.22 ^b^	55.79 ± 1.16 ^a^	34.44 ± 2.49 ^a^
30% FC (D2)	T0	51.14 ± 6.75 ^a^	1.46 ± 0.03 ^b^	0.17 ± 0.02 ^a^	0.34 ± 0.02 ^b^	1.97 ± 0.03 ^d^	61.80 ± 9.24 ^d^	13.05 ± 0.50 ^c^	4.86 ± 0.20 ^a^	4.77 ± 0.14 ^a^	31.29 ± 1.09 ^a^	65.93 ± 1.11 ^a^	2.75 ± 0.08 ^b^	33.28 ± 0.96 ^b^	20.36 ± 2.23 ^b^
	T1	2.71 ± 0.01 ^b^	1.35 ± 0.09 ^b^	0.40 ± 0.14 ^a^	0.31 ± 0.04 ^b^	2.06 ± 0.02 ^c^	290.57 ± 6.12 ^b^	15.09 ± 0.43 ^c^	2.65 ± 0.13 ^b^	4.94 ± 0.15 ^a^	33.40 ± 0.94 ^a^	53.63 ± 4.32 ^b^	3.35 ± 0.12 ^b^	57.01 ± 1.85 ^a^	32.85 ± 0.75 ^a^
	T2	6.26 ± 0.07 ^b^	2.45 ± 0.33 ^a^	0.40 ± 0.12 ^a^	0.59 ± 0.06 ^a^	3.44 ± 0.50 ^a^	118.10 ± 15.64 ^c^	24.44 ± 0.60 ^a^	5.20 ± 0.06 ^a^	4.75 ± 0.06 ^a^	31.98 ± 1.83 ^a^	24.62 ± 1.46 ^c^	9.17 ± 1.66 ^a^	60.63 ± 1.18 ^a^	29.60 ± 1.14 ^a^
	T3	4.46 ± 0.10 ^b^	2.19 ± 0.10 ^a^	0.17 ± 0.07 ^a^	0.54 ± 0.02 ^a^	2.90 ± 0.18 ^b^	393.01 ± 18.22 ^a^	18.62 ± 1.14 ^b^	5.01 ± 0.66 ^a^	5.05 ± 0.85 ^a^	32.98 ± 1.22 ^a^	54.67 ± 1.27 ^b^	2.93 ± 0.09 ^b^	59.10 ± 1.25 ^a^	29.75 ± 0.52 ^a^
**Two-Way ANOVA**															
Interaction		****	ns	ns	*	ns	****	****	****	***	****	****	****	**	***
Treatment		****	**	**	****	***	****	****	****	****	***	****	****	****	****
Drought regimes		****	****	ns	****	***	****	****	****	*	****	****	**	****	****

The results were recorded as the mean of triplicates ± standard error (S.E). Different superscript letters refer to significant differences (*p* ≤ 0.05) (Duncan’s multiple range test), ****, ***, **, * denote significant at *p* ≤ 0.0001, *p* ≤ 0.001, *p* ≤ 0.01, *p* ≤ 0.05, respectively, and ns donates non-significant difference. EL (electrolyte leakage), Chl a (chlorophyll a), Chl b (chlorophyll b), and CAT (catalase), POD (peroxidase), and PPO (polyphenol oxidase).

**Table 4 ijms-25-10954-t004:** Agronomic attributes of 22 yield traits of 102 DAS of *T. aestivum* under different water regimes.

Treatment		PL (cm)	SHL (cm)	RL (cm)	SH/R	L. No	Flag Leaf Growth Parameter						RW (cm)	R. No	FWSH (g)	DWSH (g)	FWR (g)	DWR (g)	WCSH (%)	WCR (%)
	Parameter						FLL (cm)	FLW (cm)	FLL/FLW	FLA (cm^2^)	FLP (cm)	FLAn (°)								
100% FC (D0)	T0	46.13 ± 0.64 ^b^	37.65 ± 0.27 ^b^	9.24 ± 1.24 ^b^	4.52 ± 0.54 ^a^	8.33 ± 0.33 ^a^	9.38 ± 0.24 ^b^	0.67 ± 0.07 ^b^	14.23 ± 1.55 ^c^	6.84 ± 0.19 ^b^	20.49 ± 0.61 ^b^	17.08 ± 1.27 ^c^	0.12 ± 0.03 ^b^	15.33 ± 1.76 ^b^	1.40 ± 0.10 ^a^	1.24 ± 0.09 ^a^	0.09 ± 0.02 ^a^	0.09 ± 0.02 ^a^	11.36 ± 0.86 ^b^	7.63 ± 2.89 ^d^
	T1	51.77 ± 1.02 ^a^	42.55 ± 0.62 ^a^	11.03 ± 0.27 ^a^	4.57 ± 0.56 ^a^	9.33 ± 0.88 ^a^	15.22 ± 1.22 ^a^	0.84 ± 0.05 ^a^	18.21 ± 1.27 ^a^	17.30 ± 1.39 ^a^	32.97 ± 2.13 ^a^	17.41 ± 1.51 ^c^	0.18 ± 0.00 ^a^	20.33 ± 0.88 ^a^	1.37 ± 0.06 ^b^	1.15 ± 0.07 ^a^	0.07 ± 0.00 ^a^	0.07 ± 0.00 ^a^	16.12 ± 2.93 ^a^	13.80 ± 4.54 ^c^
	T2	42.10 ± 1.27 ^b^	34.82 ± 1.06 ^b^	7.40 ± 0.21 ^c^	4.65 ± 0.10 ^a^	8.67 ± 0.33 ^a^	9.45 ± 0.27 ^b^	0.56 ± 0.03 ^b^	16.79 ± 0.49 ^b^	6.05 ± 0.55 ^c^	20.05 ± 0.49 ^b^	22.38 ± 0.96 ^b^	0.08 ± 0.02 ^c^	9.67 ± 1.20 ^d^	0.71 ± 0.10 ^c^	0.68 ± 0.10 ^b^	0.03 ± 0.01 ^a^	0.03 ± 0.01 ^b^	4.20 ± 0.40 ^c^	22.62 ± 4.29 ^b^
	T3	33.99 ± 2.33 ^c^	27.88 ± 1.35 ^c^	9.13 ± 0.84 ^b^	4.64 ± 0.72 ^a^	8.00 ± 0.58 ^a^	7.15 ± 0.40 ^c^	0.54 ± 0.04 ^b^	13.20 ± 0.27 ^d^	3.92 ± 0.56 ^d^	15.40 ± 0.97 ^c^	32.11 ± 0.93 ^a^	0.11 ± 0.03 ^b^	10.33 ± 2.03 ^c^	0.54 ± 0.11 ^c^	0.47 ± 0.08 ^b^	0.12 ± 0.09 ^a^	0.03 ± 0.00 ^b^	13.12 ± 3.32 ^a^	37.78 ± 2.22 ^a^
50% FC (D1)	T0	47.91 ± 0.90 ^b^	38.83 ± 0.71 ^b^	11.33 ± 0.78 ^b^	4.42 ± 0.52 ^a^	7.67 ± 1.76 ^c^	12.44 ± 0.71 ^b^	0.81 ± 0.09 ^a^	16.01 ± 2.86 ^a^	10.71 ± 0.54 ^b^	26.58 ± 1.33 ^a^	19.84 ± 0.57 ^b^	0.15 ± 0.02 ^b^	20.67 ± 3.18 ^b^	0.76 ± 0.08 ^b^	0.67 ± 0.08 ^c^	0.10 ± 0.01 ^b^	0.06 ± 0.01 ^b^	12.48 ± 0.94 ^a^	5.16 ± 2.60 ^a^
	T1	59.72 ± 0.53 ^a^	49.29 ± 0.49 ^a^	12.48 ± 1.13 ^a^	4.66 ± 0.25 ^a^	7.33 ± 0.88 ^c^	13.08 ± 0.38 ^a^	0.88 ± 0.07 ^a^	15.04 ± 1.19 ^a^	12.22 ± 0.55 ^a^	28.01 ± 0.93 ^a^	26.19 ± 0.42 ^a^	0.21 ± 0.02 ^a^	18.33 ± 3.18 ^b^	0.67 ± 0.13 ^b^	0.59 ± 0.13 ^c^	0.08 ± 0.00 ^a^	0.04 ± 0.01 ^b^	13.30 ± 3.47 ^a^	18.01 ± 2.63 ^a^
	T2	43.77 ± 1.82 ^b^	36.08 ± 1.47 ^b^	9.27 ± 1.08 ^c^	4.68 ± 0.20 ^a^	9.00 ± 1.53 ^b^	11.22 ± 0.52 ^b^	0.67 ± 0.05 ^a^	16.81 ± 1.44 ^a^	8.58 ± 0.76 ^c^	24.15 ± 1.08 ^b^	15.39 ± 0.81 ^c^	0.12 ± 0.03 ^c^	14.33 ± 1.86 ^c^	0.86 ± 0.02 ^b^	0.76 ± 0.06 ^b^	0.04 ± 0.01 ^b^	0.07 ± 0.01 ^b^	11.45 ± 4.65 ^a^	3.33 ± 3.33 ^a^
	T3	46.2 ± 1.35 ^b^	38.82 ± 0.86 ^b^	8.52 ± 0.5 ^c^	5.05 ± 0.24 ^a^	12.67 ± 0.88 ^a^	10.35 ± 0.79 ^c^	0.68 ± 0.06 ^a^	15.32 ± 1.02 ^a^	7.14 ± 1.10 ^d^	22.16 ± 1.53 ^b^	16.26 ± 0.47 ^c^	0.18 ± 0.02 ^b^	24.33 ± 2.85 ^a^	1.41 ± 0.28 ^a^	1.20 ± 0.21 ^a^	0.05 ± 0.00 ^b^	0.10 ± 0.01 ^a^	13.87 ± 2.14 ^a^	17.48 ± 7.20 ^a^
30% FC (D2)	T0	43.50 ± 0.29 ^c^	36.77 ± 1.34 ^a^	7.04 ± 1.69 ^a^	5.82 ± 1.13 ^a^	8.67 ± 0.88 ^a^	10.31 ± 0.31 ^a^	0.51 ± 0.01 ^d^	20.19 ± 0.60 ^a^	6.62 ± 0.26 ^c^	21.92 ± 0.50 ^a^	40.89 ± 6.31 ^b^	0.13 ± 0.03 ^b^	16.67 ± 0.88 ^a^	0.93 ± 0.02 ^c^	0.81 ± 0.04 ^c^	0.09 ± 0.01 ^a^	0.08 ± 0.01 ^a^	13.34 ± 2.51 ^a^	9.44 ± 0.56 ^b^
	T1	49.57 ± 1.43 ^a^	40.65 ± 1.76 ^a^	9.41 ± 0.60 ^a^	4.63 ± 0.51 ^a^	6.00 ± 1.15 ^a^	12.62 ± 1.20 ^a^	0.61 ± 0.01 ^c^	20.55 ± 1.96 ^a^	10.09 ± 0.96 ^b^	26.49 ± 2.26 ^a^	37.47 ± 4.32 ^a^	0.20 ± 0.03 ^a^	19.00 ± 0.58 ^a^	1.56 ± 0.21 ^a^	1.27 ± 0.15 ^a^	0.16 ± 0.05 ^a^	0.09 ± 0.01 ^a^	18.54 ± 1.65 ^a^	40.95 ± 8.38 ^a^
	T2	46.94 ± 1.12 ^b^	38.04 ± 0.61 ^a^	8.99 ± 1.33 ^a^	4.22 ± 0.69 ^a^	6.67 ± 0.33 ^a^	12.61 ± 1.74 ^a^	0.70 ± 0.10 ^b^	18.07 ± 1.64 ^a^	11.95 ± 2.10 ^a^	27.12 ± 3.54 ^a^	9.08 ± 0.93 ^c^	0.11 ± 0.01 ^b^	10.33 ± 1.33 ^b^	0.76 ± 0.11 ^d^	0.67 ± 0.10 ^d^	0.08 ± 0.00 ^a^	0.05 ± 0.00 ^b^	11.67 ± 1.33 ^a^	38.80 ± 0.72 ^a^
	T3	46.20 ± 0.95 ^b^	40.19 ± 0.72 ^a^	6.71 ± 0.20 ^a^	6.85 ± 0.66 ^a^	8.00 ± 0.58 ^a^	11.88 ± 0.88 ^a^	0.92 ± 0.05 ^a^	12.95 ± 0.34 ^b^	11.97 ± 1.96 ^a^	25.59 ± 2.04 ^a^	55.94 ± 6.41 ^d^	0.12 ± 0.01 ^b^	15.67 ± 1.86 ^a^	1.37 ± 0.17 ^b^	1.11 ± 0.13 ^b^	0.15 ± 0.00 ^a^	0.09 ± 0.00 ^a^	17.97 ± 4.26 ^a^	38.52 ± 1.70 ^a^
**Two-Way ANOVA**																				
Interaction		****	****	ns	*	*	**	***	ns	****	***	****	ns	*	****	****	****	****	*	*
Treatment		****	****	**	ns	ns	****	ns	**	****	****	****	***	***	***	ns	*	ns	*	ns
Drought regimes		****	****	**	**	*	*	ns	ns	ns	*	****	*	**	ns	ns	ns	**	***	****

The results were recorded as the mean of triplicates ± standard error (S.E). Different superscript letters refer to significant differences (*p* ≤ 0.05) (Duncan’s multiple range test), ****, ***, **, * denote significant at *p* ≤ 0.0001, *p* ≤ 0.001, *p* ≤ 0.01, *p* ≤ 0.05, respectively, and ns donates non-significant difference. PL (plant length), SHL (shoot length), RL (root length), SH/R ratio (shoot to root length ratio), L. no (leaf number), FLL (flag leaf length), FLW (flag leaf width), FLA (flag leaf area), and FLP (flag leaf perimeter), R. no (root number), RW (root width), FWSH (shoot fresh weight), FWR (root fresh weight), DWSH (shoot dry weight), DWR (root dry weight), WCSH (shoot water content), WCR (root water content).

**Table 5 ijms-25-10954-t005:** The kernel or grain parameters of 102 DAS of *T. aestivum* under different water regimes.

Treatment		Weight of 20 Grains	Grain Width (cm)	Grain Perimeter (cm)	Grain Area (cm^2^)	L (cm)	W (cm)	Aspect Ratio (AR)	Circ.	Round	Feret	Fertile Spikelets (FS)	Sterile Spikelets (SS)	No. of Spikelets/Spike (S. No.)
	Parameter													
100% FC (D0)	T0	0.17 ± 0.01 ^b^	0.20 ± 0.00 ^a^	1.50 ± 0.09 ^a^	0.11 ± 0.01 ^a^	0.65 ± 0.05 ^a^	0.22 ± 0.01 ^a^	2.30 ± 0.31 ^a^	0.59 ± 0.03 ^a^	0.34 ± 0.04 ^a^	0.63 ± 0.02 ^a^	16.33 ± 2.85 ^b^	21.00 ± 2.65 ^a^	38.00 ± 5.00 ^a^
	T1	0.31 ± 0.01 ^a^	0.21 ± 0.01 ^a^	1.46 ± 0.04 ^a^	0.13 ± 0.02 ^a^	0.65 ± 0.03 ^a^	0.26 ± 0.03 ^a^	2.59 ± 0.26 ^a^	0.65 ± 0.02 ^a^	0.39 ± 0.04 ^a^	0.66 ± 0.02 ^a^	32.00 ± 5.29 ^a^	11.00 ± 3.79 ^c^	43.00 ± 2.08 ^a^
	T2	0.02 ± 0.00 ^d^	0.20 ± 0.00 ^a^	1.40 ± 0.03 ^a^	0.13 ± 0.01 ^a^	0.64 ± 0.01 ^a^	0.25 ± 0.01 ^a^	2.49 ± 0.24 ^a^	0.64 ± 0.03 ^a^	0.41 ± 0.04 ^a^	0.62 ± 0.00 ^a^	1.00 ± 0.58 ^c^	21.00 ± 2.65 ^a^	21.00 ± 2.65 ^b^
	T3	0.13 ± 0.00 ^c^	0.18 ± 0.02 ^a^	1.40 ± 0.01 ^a^	0.10 ± 0.00 ^a^	0.62 ± 0.02 ^a^	0.20 ± 0.01 ^a^	3.17 ± 0.25 ^a^	0.58 ± 0.02 ^a^	0.32 ± 0.02 ^a^	0.62 ± 0.00 ^a^	8.00 ± 3.21 ^d^	17.00 ± 1.73 ^b^	25.00 ± 1.53 ^b^
50% FC (D1)	T0	0.31 ± 0.01 ^a^	0.22 ± 0.04 ^a^	1.43 ± 0.10 ^a^	0.13 ± 0.01 ^b^	0.61 ± 0.03 ^a^	0.26 ± 0.01 ^b^	2.34 ± 0.04 ^a^	0.70 ± 0.03 ^a^	0.43 ± 0.01 ^b^	0.65 ± 0.06 ^a^	15.00 ± 4.51 ^a^	19.00 ± 4.00 ^a^	34.00 ± 0.58 ^a^
	T1	0.24 ± 0.01 ^b^	0.22 ± 0.00 ^a^	1.43 ± 0.03 ^a^	0.11 ± 0.00 ^b^	0.59 ± 0.02 ^a^	0.23 ± 0.00 ^b^	2.61 ± 0.13 ^a^	0.67 ± 0.01 ^a^	0.39 ± 0.02 ^b^	0.61 ± 0.02 ^a^	26.00 ± 2.31 ^a^	14.33 ± 1.86 ^a^	37.00 ± 3.61 ^a^
	T2	0.23 ± 0.01 ^b^	0.26 ± 0.00 ^a^	1.49 ± 0.02 ^a^	0.16 ± 0.01 ^a^	0.61 ± 0.03 ^a^	0.34 ± 0.02 ^a^	1.82 ± 0.18 ^b^	0.72 ± 0.02 ^a^	0.56 ± 0.06 ^a^	0.62 ± 0.02 ^a^	18.67 ± 1.20 ^a^	13.00 ± 1.15 ^a^	31.67 ± 2.19 ^a^
	T3	0.16 ± 0.00 ^c^	0.20 ± 0.00 ^a^	1.45 ± 0.09 ^a^	0.11 ± 0.01 ^b^	0.61 ± 0.04 ^a^	0.23 ± 0.01 ^b^	2.64 ± 0.16 ^a^	0.65 ± 0.03 ^a^	0.38 ± 0.02 ^b^	0.58 ± 0.03 ^a^	20.00 ± 3.61 ^a^	15.00 ± 1.15 ^a^	35.00 ± 2.52 ^a^
30% FC (D2)	T0	0.21 ± 0.01 ^d^	0.19 ± 0.01 ^b^	1.24 ± 0.02 ^c^	0.08 ± 0.00 ^d^	0.51 ± 0.02 ^b^	0.20 ± 0.01 ^c^	2.63 ± 0.17 ^a^	0.67 ± 0.02 ^b^	0.38 ± 0.02 ^b^	0.52 ± 0.00 ^b^	19.33 ± 3.38 ^b^	14.67 ± 2.73 ^a^	34.00 ± 4.04 ^c^
	T1	0.25 ± 0.01 ^b^	0.21 ± 0.02 ^b^	1.38 ± 0.02 ^b^	0.11 ± 0.02 ^c^	0.60 ± 0.03 ^a^	0.23 ± 0.03 ^c^	2.66 ± 0.19 ^a^	0.65 ± 0.01 ^c^	0.38 ± 0.03 ^b^	0.61 ± 0.02 ^a^	31.67 ± 0.88 ^a^	11.67 ± 1.20 ^a^	43.33 ± 1.45 ^a^
	T2	0.27 ± 0.01 ^a^	0.28 ± 0.10 ^a^	1.57 ± 0.01 ^a^	0.14 ± 0.00 ^a^	0.63 ± 0.02 ^a^	0.29 ± 0.01 ^b^	2.16 ± 0.15 ^a^	0.76 ± 0.03 ^a^	0.47 ± 0.03 ^a^	0.64 ± 0.02 ^a^	12.00 ± 6.24 ^c^	16.67 ± 7.06 ^a^	28.67 ± 0.88 ^d^
	T3	0.25 ± 0.01 ^c^	0.22 ± 0.00 ^b^	1.54 ± 0.08 ^a^	0.13 ± 0.00 ^b^	0.66 ± 0.02 ^a^	0.24 ± 0.01 ^a^	2.71 ± 0.17 ^a^	0.64 ± 0.04 ^b^	0.37 ± 0.02 ^c^	0.63 ± 0.01 ^a^	27.00 ± 2.52 ^a^	11.67 ± 1.33 ^a^	39.33 ± 2.60 ^b^
**Two-Way ANOVA**														
Interaction		****	ns	*	**	ns	*	ns	ns	ns	*	*	ns	*
Treatment		****	**	ns	***	ns	****	**	**	***	ns	****	ns	****
Drought regimes		****	*	ns	ns	ns	*	*	***	*	ns	**	ns	ns

The results were recorded as the mean of triplicates ± standard error (S.E). Different superscript letters refer to significant differences (*p* ≤ 0.05) (Duncan’s multiple range test), ****, ***, **, * denote significant at *p* ≤ 0.0001, *p* ≤ 0.001, *p* ≤ 0.01, *p* ≤ 0.05, respectively, and ns donates non-significant difference; L (length of grain major axis); W (length of grain minor axis); AR (aspect ratio); Circ. (circulatory); Round (roundness); Feret (Feret diameter). S. No (no. of spikelets/spike); FS (fertile spikelets); SS (sterile spikelets).

**Table 6 ijms-25-10954-t006:** Drought tolerance indices of 102 DAS of *T. aestivum* under different water regimes.

Treatment		MP	GMP	TOL	SI	SSI	STI	HARM	YSI	RDI	DRI	YRR	YI
	Parameter												
FC (50%) (D1)	T0	20.00	19.38	2.00	0.12	−4.98	1.24	18.80	0.98	0.76	0.92	0.02	0.84
	T1	12.67	12.34	−3.33		−5.60	0.55	12.04	1.60	1.24	0.98	−0.60	0.64
	T2	17.00	16.40	8.00		−4.64	0.88	15.82	0.64	0.50	0.38	0.36	0.58
	T3	16.00	15.87	2.00		−4.91	0.83	15.75	0.91	0.71	0.62	0.09	0.67
FC (30%) (D2)	T0	20.17	19.85	1.67	−0.29	−4.96	1.31	19.54	0.96	0.75	0.88	0.04	0.86
	T1	21.33	18.14	−20.67		−7.69	1.15	15.65	3.69	2.87	5.12	−2.69	1.41
	T2	16.50	12.88	9.00		−4.60	0.81	12.60	0.60	0.47	0.51	0.40	0.53
	T3	22.00	21.31	−10.00		−5.63	1.50	20.65	1.52	1.27	2.00	−0.63	1.31

Mean productivity (MP); geometric productivity (GMP); tolerance index (TOL); stress susceptibility index (SSI); stress tolerance index (STI); harmonic mean of yield (HARM); yield stability index (YSI); relative drought index (RDI); drought resistance index (DRI); yield reduction ratio (YRR), and yield index (YI).

**Table 7 ijms-25-10954-t007:** The sequences of primer pairs used for the gene expression quantitative real-time PCR (qRT-PCR) of five drought-responsive genes.

Primer	Primer (5′→3′) Forward Sequence	Primer (5′→3′) Reverse Sequence	Reference
*TaZFP34*	ACGGCGATCAGTGGGTGT	GACGAACAGCTCGAGCAAGA	Chang, et al. [80]
*TaWRKY1*	ATGTGGGAAAATGGTAAAA	CTATCTTTCCTTTCTTTGC	Ding et al. [128]
*TaMAPK3*	CTTTAACCCGCTGCAGAGGA	GTCAAAGGAGAAGGGGTCCG	Dudziak et al. [129]
*TaLEA*	GACAACACCATCACCACCAAGGACA	TAATACAGAACCGGACACGAGGAGT	Itam et al. [74]
*TaP5CS1*	TGGCCTTGTGAAAAGCAAAGA	GCCTGTTACTGCCTCTTGGA	Itam et al. [74]
*TaActin*	CCTCTCTGCGCCAATCGT	TCAGCCGAGCGGGAAATTGT	Itam et al. [74]

## Data Availability

Data are contained within the article and Appendix A.

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
