# Peer review of "Stress-Responsive Gene Expression, Metabolic, Physiological, and Agronomic Responses by Consortium Nano-Silica with Trichoderma against Drought Stress in Bread Wheat"

_ijms, 2024, doi:10.3390/ijms252010954_

Round 1

Reviewer 1 Report

Comments and Suggestions for Authors

The paper studies the responses of a wheat treated with nanoparticles under the drought stress, which is of great significance in the relevant research field. However, all the sections in the entire essay are too redundant. Sentences are too long (over 4 lines) which reduces the readability and hinder the understanding. Too many grammatical errors. Overall, the language should be polished throughout the manuscript to make it more concise and accurate. The following are several specific comments:

1.        The title should be more focused, covering critical information, for example, SiO2 NPs.

2.        The Introduction section is too redundant and the logic is not clear. The focus should be on wheat, not all the crops. And the basic common-sense concepts/knowledge should be deleted. Hypotheses of the study instead results should be provided at the end of section.

3.        Too many inappropriate abbreviations, increasing the memory burden of the readers and hindering their understanding, such as DAS, MP, YSI, SSI, STI, D1,D2, EL, et al.

4.        Inappropriate use of brackets should be corrected, for example, Line 19, et al.

5.        Incorrect use of %, which should not be shared by several numbers.

6.        The sections of methods, results, and discussion should be more concise and integrated.

7.        Too many of the references are outdated.

Comments on the Quality of English Language

Sentences are too long (over 4 lines) and the logic is not clear. Grammatical errors should be checked and corrected throughout the manuscript.

Author Response

COMMENTS TO BOTH EDITOR AND REVIEWERS:

We introduced an updated bioinformatics approach throughout the text, highlighted in green, from the abstract to the conclusion, and the supplemental materials. We would like to get your attention and confirm our idea.

COMMENTS FROM REVIEWERS

Thank you for your positive comments. We would like to acknowledge the reviewer’s efforts and positive overall evaluation for our manuscript. We also appreciate all reviewers for taking their time to evaluate our manuscript. All thoughtful comments have been fully addressed in our revision. The corrections of the reviewers were made and highlighted in (turquois, and yellow) colours as reviewer1, and 2.

Reviewer #1:

- Reviewer comment: The paper studies the responses of a wheat treated with nanoparticles under the drought stress, which is of great significance in the relevant research field. However, all the sections in the entire essay are too redundant. Sentences are too long (over 4 lines) which reduces the readability and hinder the understanding. Too many grammatical errors. Overall, the language should be polished throughout the manuscript to make it more concise and accurate. The following are several specific comments:

Authors response: We would like to send a sincere thanks to our reviewer for thoughtful critiques of our manuscript. We agreed with all comments and suggestions of our reviewer. We found them quite useful as we improve our paper. We checked the long sentences and improved it. Also, we checked the English language of all text and corrected the style and typos.

- Reviewer comment 1.   The title should be more focused, covering critical information, for example, SiO2 NPs.

Authors response: Thank of your suggestion. We changed the title to be more focused.

- Reviewer comment 2.  The Introduction section is too redundant and the logic is not clear. The focus should be on wheat, not all the crops. And the basic common-sense concepts/knowledge should be deleted. Hypotheses of the study instead results should be provided at the end of section.

Authors response: Thanks for your comment. We justified the introduction section by removing some sentences and a new paragraph related to the molecular approaches.

- Reviewer comment 3.  Too many inappropriate abbreviations, increasing the memory burden of the readers and hindering their understanding, such as DAS, MP, YSI, SSI, STI, D1, D2, EL, et al.

Authors response: Thanks for your comment. We mentioned all in the material and methods section but if we wrote them in full name, it will take a great space.

- Reviewer comment 4. Inappropriate use of brackets should be corrected, for example, Line 19, et al.

Authors response: Thanks for your comment. We corrected it.

- Reviewer comment 5.  Incorrect use of %, which should not be shared by several numbers.

Authors response: Thanks for your comment. We corrected it.

- Reviewer comment 6. The sections of methods, results, and discussion should be more concise and integrated.

 Authors response: Thanks for your comment. We summarized the text in all sections.

- Reviewer comment 7.  Too many of the references are outdated.

Authors response: Thanks for your comment. We removed some references and added more recent ones such as:

  • Makhadmeh, I.M.; Thabet, S.G.; Ali, M.; Alabbadi, B.; Albalasmeh, A.; Alqudah, A.M. Exploring genetic variation among Jordanian Solanum lycopersicon L. landraces and their performance under salt stress using SSR markers. Journal of Genetic Engineering and Biotechnology 2022, 20, 45.
  • Makhadmeh, I.; Albalasmeh, A.A.; Ali, M.; Thabet, S.G.; Darabseh, W.A.; Jaradat, S.; Alqudah, A.M. Molecular characterization of tomato (Solanum lycopersicum L.) accessions under drought stress. Horticulturae 2022, 8, 600.
  • Abdelhameed, A.A.; Ali, M.; Darwish, D.B.E.; AlShaqhaa, M.A.; Selim, D.A.-F.H.; Nagah, A.; Zayed, M. Induced genetic diversity through mutagenesis in wheat gene pool and significant use of SCoT markers to underpin key agronomic traits. BMC Plant Biology 2024, 24, 673.

Reviewer 2 Report

Comments and Suggestions for Authors

The main question addressed by the research is the investigated a show to prevent drought stress using a single and combined application of PGPM and nSiO2 in wheat.

Different case studies, similar solutions compared with other published

material.

General comments: the manuscript resulted robust and well written. Only minor queries need to be addressed. 

Methodology are clear and well reported 

The conclusions are consistent with the evidence and arguments presented and they address the main question posed.

The references are appropriate.

Detailed comment:

Figure 4 is flattened and results difficult to read.

Figures 5-6 need to be improved in term of resolution.  

Figure 7 is flattened and results difficult to read.

Figure 8 is small and flattened; it results difficult to read.

Table 7: please include also geneID of those considered in qPCR analysis.

Author Response

COMMENTS TO BOTH EDITOR AND REVIEWERS:

We introduced an updated bioinformatics approach throughout the text, highlighted in green, from the abstract to the conclusion, and the supplemental materials. We would like to get your attention and confirm our idea.

COMMENTS FROM REVIEWERS

Thank you for your positive comments. We would like to acknowledge the reviewer’s efforts and positive overall evaluation for our manuscript. We also appreciate all reviewers for taking their time to evaluate our manuscript. All thoughtful comments have been fully addressed in our revision. The corrections of the reviewers were made and highlighted in (turquois, and yellow) colours as reviewer1, and 2.

Reviewer #2

- Reviewer comment: The main question addressed by the research is the investigated a show to prevent drought stress using a single and combined application of PGPM and nSiO2 in wheat. Different case studies, similar solutions compared with other published material.

 Authors response: Thanks for your positive comment. We would like to extend a sincere thanks to our reviewer for constructive comments on our work.

General comments: 

- Reviewer comment: the manuscript resulted robust and well written. Only minor queries need to be addressed. 

 Authors response: Thanks for your positive comment. We made all required corrections.

- Reviewer comment: Methodology are clear and well reported 

 Authors response: Thanks for your positive comment. We appreciate this great supportive comment.

- Reviewer comment: The conclusions are consistent with the evidence and arguments presented and they address the main question posed.

 Authors response: Thanks for your positive comment. We appreciate this great supportive comment, and we added a new supportive part.

- Reviewer comment: The references are appropriate.

 Authors response: Thanks for your positive comment. We also added recent ones.

Detailed comments:

- Reviewer comment: Figure 4 is flattened and results difficult to read.

 Authors response: Thanks for your comment. We modified it to be more informative.

- Reviewer comment: Figures 5-6 need to be improved in term of resolution.  

 Authors response: Thanks for your comment. We modified all figures to be clearer.

- Reviewer comment: Figure 7 is flattened and results difficult to read.

 Authors response: Thanks for your comment. We stretched it to be more informative.

- Reviewer comment: Figure 8 is small and flattened; it results difficult to read.

 Authors response: Thanks for your comment. We stretched it to be more obvious.

- Reviewer comment: Table 7: please include also gene ID of those considered in qPCR analysis.

Authors response: Thanks for your comments. We added the ID of the studied genes in the materials and methods section due to the small space in the Table. We cannot added a new column.

Thank you very much for your insightful remarks and comments that helped us enhance the manuscript.